# Continuous Semi-Implicit Models

**Longlin Yu** [* 1]  **Jiajun Zha** [* 2]  **Tong Yang** [3 4]  **Tianyu Xie** [1]  **Xiangyu Zhang** [4]  **S.-H. Gary Chan** [2]  **Cheng Zhang** [1 5]

## Abstract

Semi-implicit distributions have shown great promise in variational inference and generative modeling. Hierarchical semi-implicit models, which stack multiple semi-implicit layers, enhance the expressiveness of semi-implicit distributions and can be used to accelerate diffusion models given pretrained score networks. However, their sequential training often suffers from slow convergence. In this paper, we introduce CoSIM, a continuous semi-implicit model that extends hierarchical semi-implicit models into a continuous framework. By incorporating a continuous transition kernel, CoSIM enables efficient, simulation-free training. Furthermore, we show that CoSIM achieves consistency with a carefully designed transition kernel, offering a novel approach for multistep distillation of generative models at the distributional level. Extensive experiments on image generation demonstrate that CoSIM performs on par or better than existing diffusion model acceleration methods, achieving superior performance on FD-DINOv2.

## 1. Introduction

Semi-implicit distributions, which are constructed through the convolution of explicit conditional distributions and implicit mixing distributions, have gained significant traction in variational inference and generative modeling. Unlike traditional approximating distributions with explicit density forms, semi-implicit distributions enable a more expressive family of variational posteriors, leading to improved approx-

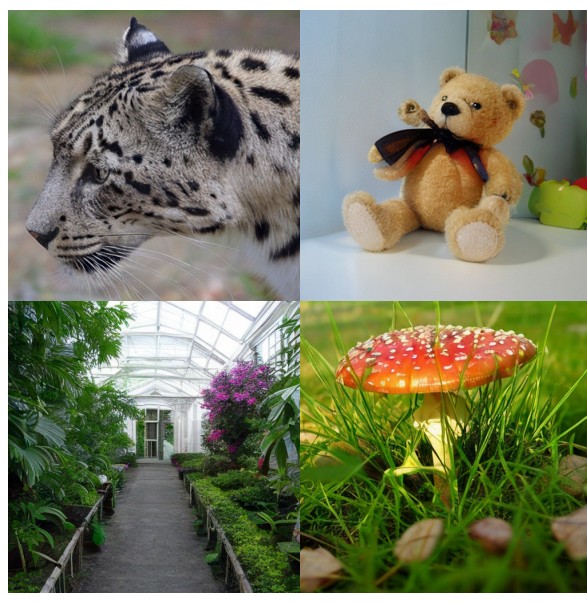

*Figure 1.* Selected generated images on Imagenet $512 \times 512$ using L model from Section 4.2.

imation accuracy (Yin & Zhou, 2018; Titsias & Ruiz, 2019; Moens et al., 2021; Yu & Zhang, 2023; Cheng et al., 2024). Beyond variational inference, semi-implicit architectures have been successfully integrated into deep generative models, including variational autoencoders (VAEs) (Kingma & Welling, 2014; Rezende et al., 2014) and diffusion models (Sohl-Dickstein et al., 2015; Song & Ermon, 2019; Ho et al., 2020; Song et al., 2021; 2020).

In VAEs, the generator typically employs a single-layer semi-implicit construction, pushing forward a simple noise through a conditional factorized Gaussian distribution parameterized by a neural network. However, despite their widespread adoption, VAE-generated samples often suffer from blurriness and fail to capture high-frequency details in the data distribution (Dosovitskiy & Brox, 2016). To address these limitations, several VAE variants have been proposed (Rezende & Mohamed, 2016; Razavi et al., 2019; Vahdat & Kautz, 2021). A prominent approach is hierarchical VAEs (Gregor et al., 2015; Ranganath et al., 2016; Sønderby et al., 2016; Kingma et al., 2017; Vahdat & Kautz, 2021), which enhances the expressiveness of the single layer vallina VAEs through the introduction of multiple latent vari-

---

[*]Equal contribution. This work was done during an internship at StepFun. [1]School of Mathematical Sciences, Peking University, Beijing, China [2]Department of Computer Science and Engineering, Hong Kong University of Science and Technology [3]School of Computer Science, Fudan University, Shanghai, China [4]Megvii Technology Inc., Beijing, China [5]Center for Statistical Science, Peking University, Beijing, China. Correspondence to: Cheng Zhang <chengzhang@math.pku.edu.cn>.

*Proceedings of the 42nd International Conference on Machine Learning*, Vancouver, Canada. PMLR 267, 2025. Copyright 2025 by the author(s).

ables, enabling multistep generation. Similarly, diffusion models have emerged as a leading framework for generating high-quality and diverse samples (Sohl-Dickstein et al., 2015; Song & Ermon, 2019; Ho et al., 2020; Song et al., 2021; 2020). These models can also be interpreted through the lens of semi-implicit distributions, where a fixed noise injection process is used to generate latent variables instead of learnable posterior distributions as in hierarchical VAEs.

Although multistep generative models are a promising direction for improving sample quality, they often involve a trade-off between generation quality and computational cost. Drawing on the connection between diffusion models, hierarchical VAEs, and semi-implicit distributions, recent works have explored distilling diffusion models using tools from semi-implicit variational inference (SIVI) (Yin et al., 2023; Luo et al., 2023; Yu et al., 2023; Zhou et al., 2024b;a). These methods can be broadly categorized into two types: (1) one-step deterministic distillation methods, which learn a direct mapping to generate samples from the target distribution (Yin et al., 2023; Luo et al., 2023; Zhou et al., 2024b;a), and (2) stochastic multi-step models, which generate more diverse samples in fewer steps. In particular, hierarchical semi-implicit variational inference (HSIVI) falls into the second category, recursively constructing the variational distribution over a fixed sequence of time points (Yu et al., 2023). While HSIVI can generate more diverse samples than one-step models, its discretized design results in slow convergence during training due to the sequential simulation process.

Drawing inspiration from continuous-time diffusion processes (Song et al., 2020), we introduce a Continuous Semi-Implicit Model (CoSIM), which extends hierarchical semi-implicit variational inference (HSIVI) into a continuous framework. By leveraging a continuous transition kernel, CoSIM generates samples of the mixing layer in a single push-forward operation, significantly accelerating the training process. The design of the continuous transition kernel shares some similarities with consistency distillation (CD) (Song et al., 2023; Kim et al., 2023; Song & Dhariwal, 2024; Geng et al., 2024; Heek et al., 2024; Lu & Song, 2024), which learn a consistency function to map noisy distributions back to clean target distributions. While Salimans et al. (2024) explore distilling multi-step diffusion models using moment-matching losses, CoSIM distinguishes itself by employing semi-implicit variational inference (SIVI) training criteria. This approach enables learning the consistency function directly at the distributional level, bypassing the need to recover the reverse process of diffusion models at the sample or moment level. As a result, CoSIM significantly reduces the number of iterations required for distillation. In experiments, we demonstrate that CoSIM achieves comparable results on the Fréchet Inception Distance (FID) (Heusel et al., 2017) while incurring lower training costs.

Furthermore, CoSIM outperforms existing methods on the FD-DINOv2 metric (Stein et al., 2024), which employs the larger DINOv2 encoder (Oquab et al., 2024) instead of the InceptionV3 encoder (Szegedy et al., 2016) to better align with human perception.

## 2. Background

### 2.1. Diffusion Models

Diffusion models (Sohl-Dickstein et al., 2015; Song & Ermon, 2019; Ho et al., 2020; Song et al., 2021; 2020) perturb the clean data to noise in the forward process and then generate the data from noise by multiple denoising steps in the backward process. The forward process can be described by a stochastic differential equation (SDE)

$$\mathrm{d}\boldsymbol{x}_s = \boldsymbol{f}(\boldsymbol{x}_s; s)\mathrm{d}s + g(s)\mathrm{d}\mathbf{B}_s, \qquad (1)$$

where $s \in [0, T]$, $T > 0$ is a fixed terminating time, $\mathbf{B}_s$ is a standard Brownian motion, and $\boldsymbol{f}(\boldsymbol{x}_s; s)$ and $g(s)$ are the drift and diffusion coefficients respectively. The starting $\boldsymbol{x}_0 \sim p(\cdot; 0)$ is the data distribution. Denote the density law of the forward process as $\{p(\cdot; s)\}_{s \in [0,T]}$. Typically, the SDE in (1) is designed as a variance preserving (VP) or variance exploding (VE) scheme (Song et al., 2020; Karras et al., 2022), and the samples of $p(\boldsymbol{x}_t | \boldsymbol{x}_0; t, 0)$ can be reparameterized as

$$\boldsymbol{x}_t = a(t)\boldsymbol{x}_0 + \sigma(t)\boldsymbol{\epsilon}, \quad t \in [0, T], \qquad (2)$$

where $\boldsymbol{\epsilon}$ is a Gaussian noise, $a(t)$ is non-increasing function, and $\sigma(t)$ is a monotonically increasing function. In the backward process, one can run the following reversed-time SDE from $T$ to $0$ to generate the samples of $p(\boldsymbol{x}_0; 0)$

$$\mathrm{d}\boldsymbol{x}_s = [\boldsymbol{f}(\boldsymbol{x}_s; s) - g^2(s)\nabla \log p(\boldsymbol{x}_s; s)]\mathrm{d}s + g(s)\mathrm{d}\bar{\mathbf{B}}_s,$$

where $\nabla \log p(\boldsymbol{x}_s; s)$ is the score function of $p(\boldsymbol{x}_s; s)$, $\bar{\mathbf{B}}_s$ is a standard Brownian motion. The score function $\nabla \log p(\boldsymbol{x}_s; s)$ is often estimated with a score model $\boldsymbol{S}_\theta(\boldsymbol{x}_s; s) \approx \nabla \log p(\boldsymbol{x}_s; s)$ via denoising score matching (Vincent, 2011; Song et al., 2020).

### 2.2. Semi-Implicit Models and Diffusion Distillation

Semi-implicit is a mixture distribution expressed as $q_\phi(\boldsymbol{x}) = \int q_\phi(\boldsymbol{x}|\boldsymbol{z})q(\boldsymbol{z})\mathrm{d}\boldsymbol{z}$, which can be used to approximate the target distribution via variational inference (Yin & Zhou, 2018; Titsias & Ruiz, 2019). The conditional layer $q_\phi(\boldsymbol{x}|\boldsymbol{z})$ is required to be explicit but the mixing layer $q(\boldsymbol{z})$ can be implicit, where $\phi$ is the variational parameter. Recent works have explored diffusion model distillation within the semi-implicit framework (Wang et al., 2023b). These methods are primarily distinguished by their training objectives: those utilizing density-based divergences (e.g., JSD and KL

divergence) (Luo et al., 2023; Wang et al., 2023a; Yin et al., 2023), and those employing score-based divergences (e.g., Fisher divergence) (Yu et al., 2023; Zhou et al., 2024b). Following SiD (Zhou et al., 2024b), which demonstrated superior performance and faster convergence in terms of FID results than Diff-Instruct (Luo et al., 2023), we adopt the score-based training objective in this work.

**Hierarchical Semi-Implicit Variational Inference** Yu et al. (2023) propose a hierarchical semi-implicit structure recursively defined from the top layer $k = K - 1$ to the base layer $k = 0$ for multistep diffusion distillation. This structure employs the conditional layer $q_\phi(\boldsymbol{x}_k | \boldsymbol{x}_{k+1}; t_k)$ and the variational prior $q(\boldsymbol{x}_T; T)$, defined by

$$q_\phi(\boldsymbol{x}_k; t_k) = \int q_\phi(\boldsymbol{x}_k | \boldsymbol{x}_{k+1}; t_k) q_\phi(\boldsymbol{x}_{k+1}; t_{k+1}) \mathrm{d}\boldsymbol{x}_{k+1}, \tag{3}$$

where $k = 0, 1, \ldots, K - 1$, $0 < t_1 < \ldots < t_K = T$ and $q_\phi(\cdot; t_K) := q(\cdot; T)$. Moreover, the forward process (1) of diffusion models naturally provides a sequence of intermediate bridge distributions $\{p(\boldsymbol{x}; t_k)\}_{k=0}^{K-1}$, which can be combined with (3) for diffusion models distillation. In the training procedure, Yu et al. (2023) introduce a joint training scheme to minimize a weighted sum of the semi-implicit variational inference (SIVI) objectives

$$\mathcal{L}_{\text{HSIVI-}f}(\phi) = \sum_{k=0}^{K-1} \beta(k) \mathcal{L}_{\text{SIVI-}f} \left( q_\phi(\cdot; t_k) \| p(\cdot; t_k) \right), \tag{4}$$

where $\beta(k) : \{0, \ldots, T - 1\} \to \mathbb{R}_+$ is a positive weighting function and $f$ represents a distance criterion. In the application to diffusion model acceleration, $f$ can be chosen as the Fisher divergence and the resulting HSIVI-SM optimizes $q_\phi(\boldsymbol{x}_t; t)$ via

$$\min_\phi \max_{\boldsymbol{v}(\cdot; t_k)} \mathcal{L}_{\text{SIVI-SM}}(\phi) = \mathbb{E}_{q_\phi(\boldsymbol{x}_{t_k}; t_k)} \Big[ \boldsymbol{v}(\boldsymbol{x}_{t_k}; t_k)^T \left[ \nabla \log p(\boldsymbol{x}_{t_k}; t_k) \right.$$

$$\left. - \nabla_{\boldsymbol{x}_{t_k}} \log q_\phi(\boldsymbol{x}_{t_k} | \boldsymbol{x}_{t_{k+1}}; t_k) \right] - \tfrac{1}{2} \| \boldsymbol{v}(\boldsymbol{x}_{t_k}; t_k) \|_2^2 \Big], \tag{5}$$

where $\boldsymbol{v}(\boldsymbol{x}_{t_k}; t_k)$ is an auxiliary vector-valued function and $q_\phi(\boldsymbol{x}_{t_k}, \boldsymbol{x}_{t_{k+1}}) = q_\phi(\boldsymbol{x}_{t_k} | \boldsymbol{x}_{t_{k+1}}; t_k) q_\phi(\boldsymbol{x}_{t_{k+1}}; t_{k+1})$.

**Score Identity Distillation** For the optimization of $\phi$ in (5), the coefficient $\frac{1}{2}$ is not a unique choice. Zhou et al. (2024b) introduced Score Identity Distillation (SiD), a one-step distillation method for diffusion models. Assuming the variational semi-implicit distribution $\tilde{q}_\phi(\boldsymbol{x}_t, \boldsymbol{z}; t) = p(\boldsymbol{x}_t | \boldsymbol{G}_\phi(\boldsymbol{z}); t, 0) \tilde{q}(\boldsymbol{z})$, where $\boldsymbol{G}_\phi$ is a learnable neural network generator mapping from a simple distribution $\tilde{q}(\boldsymbol{z})$ to data distribution and the conditional layer $p(\cdot | \cdot; t, 0)$ follows the definition in (2). SiD employs a fused loss function, which can be viewed as the training objective in SIVI objec-

tives

$$\min_\phi \quad \mathcal{L}_{\text{SiD}}(\phi) = \mathbb{E}_{\tilde{q}_\phi(\boldsymbol{x}_t, \boldsymbol{z}; t)} \Big[ \boldsymbol{v}(\boldsymbol{x}_t; t)^T \left[ \nabla \log p(\boldsymbol{x}_s; s) - \right.$$

$$\left. \nabla_{\boldsymbol{x}_t} \log \tilde{q}_\phi(\boldsymbol{x}_t | \boldsymbol{z}; t) \right] - \alpha \| \boldsymbol{v}(\boldsymbol{x}_t, t) \|_2^2 \Big], \tag{6}$$

where $\alpha \in \mathbb{R}^+$ is a given hyperparameter used to balance the cross term and squared term in (6). The lower-level optimization problem for $\boldsymbol{v}(\boldsymbol{x}_t; t)$ remains consistent with the maximization of $\mathcal{L}_{\text{SIVI-SM}}(\phi)$. Empirically, SiD with $\alpha > \frac{1}{2}$ demonstrates significantly better performance than both the baseline of $\alpha = \frac{1}{2}$ and density-based distillation methods in one-step image generation tasks (Zhou et al., 2024b).

# 3. Continuous Semi-Implicit Models

Unlike one-step generation models, stochastic multi-step approaches such as HSIVI provide a systematic framework for progressive image quality enhancement while preserving sample diversity. However, HSIVI suffers from slow convergence in its training process. Inspired by the evolution of noise conditional score network models (NCSN) (Song & Ermon, 2019) to continuous-time score-based generative models (Song et al., 2020), extending the sequential training to a continuous-time training framework promises to enhance the training efficiency. Following Yu et al. (2023), we extend the hierarchical semi-implicit model with a fixed number of layers to a continuous framework. Our goal is to learn a transition kernel $q_{\text{trans}}(\boldsymbol{x}_s | \boldsymbol{x}_t; s, t)$ that maps the distribution $p(\boldsymbol{x}_t; t)$ to $p(\boldsymbol{x}_s; s)$ for $0 \leq s < t \leq T$ as a continuous generalization of the conditional layer $q(\boldsymbol{x}_{t_k} | \boldsymbol{x}_{t_{k+1}}; t_k)$ in HSIVI. In the context of diffusion models, we assume $p(\cdot; t)$ follows the density of the forward process of (1).

## 3.1. Continuous Transition Kernel

To construct the density of variational distribution at timestep $s$, we denote the marginal distribution $q(\boldsymbol{x}_s; s, t)$ as follows

$$q(\boldsymbol{x}_s; s, t) = \int q_{\text{trans}}(\boldsymbol{x}_s | \boldsymbol{x}_t; s, t) q_{\text{mix}}(\boldsymbol{x}_t; t) \mathrm{d}\boldsymbol{x}_t, \tag{7}$$

where $q_{\text{mix}}(\boldsymbol{x}_t; t)$ is chosen as a role of the mixing distribution in semi-implicit variational distribution. Within the paradigm of hierarchical semi-implicit distribution (Yu et al., 2023), $q_{\text{mix}}(\boldsymbol{x}_{t_k}; t_k)$ is obtained by progressively sampling from the conditional layers $\{q_\phi(\boldsymbol{x}_{t_i} | \boldsymbol{x}_{t_{i+1}}; t_i)\}_{i \geq k}$ as indicated in (3). Instead, we construct $q_{\text{mix}}(\cdot; t)$ through a single push-forward operation using the conditional distribution $p(\cdot | \cdot; t, 0)$ defined in (2)

$$q_{\text{mix}}(\boldsymbol{x}_t; t) = \int p(\boldsymbol{x}_t | \boldsymbol{x}_0; t, 0) p(\boldsymbol{x}_0; 0) \mathrm{d}\boldsymbol{x}_0, \tag{8}$$

which allows us to sample $\boldsymbol{x}_t \sim q_{\text{mix}}(\boldsymbol{x}_t; t)$ instantaneously. With the continuous timesteps $t$ and $s$, we can train the tran-

sition kernel $q_{\text{trans}}(\boldsymbol{x}_s|\boldsymbol{x}_t; s, t)$ via the following continuous generalization of (4)

$$\mathcal{L}_{\text{CSI-}f}(q) = \int_0^T \pi(s, t)\mathcal{L}_{\text{SIVI-}f}(q(\boldsymbol{x}_s; s, t)\|p(\boldsymbol{x}_s; s))\, \mathrm{d}t\mathrm{d}s, \quad (9)$$

where $\pi(s, t) = \pi(s)\pi(t|s)$ is a joint time schedule. A detailed discussion of the design principles for $\pi(s)\pi(t|s)$ is presented in Appendix A.1. Note that the marginal distribution $q(\boldsymbol{x}_s; s, t)$ in (7) depends on $t$. Thus models of the continuous transition kernel $q_{\text{trans}}(\boldsymbol{x}_s|\boldsymbol{x}_t; s, t)$ will lead to consistency, as it convert $q_{\text{mix}}(\boldsymbol{x}_t; t)$ to the same distribution $p(\boldsymbol{x}_s; s)$ for all $t \in (s, T]$. We call it a continuous semi-implicit model (CoSIM).

### 3.2. The Training Objectives

Let $q_\phi(\boldsymbol{x}_s; s, t)$ be the corresponding variational distribution for the parameterized transition kernel $q_\phi(\boldsymbol{x}_s|\boldsymbol{x}_t; s, t)$. We adopt the score matching objective $\mathcal{L}_{\text{SIVI-SM}}$ introduced in Section 2.2 for training. More specifically, we parameterize the auxiliary vector-valued function as $\boldsymbol{v}_\psi(\boldsymbol{x}_s; s, t) := \nabla \log p(\boldsymbol{x}_s; s) - \boldsymbol{f}_\psi(\boldsymbol{x}_s; s, t)$ and reformulate the optimization into two stages

$$\min_\phi \; \mathbb{E}_{q_\phi(\boldsymbol{x}_s, \boldsymbol{x}_t; s, t)}\Big[\boldsymbol{v}_\psi(\boldsymbol{x}_s; s, t)^T\left[\nabla \log p(\boldsymbol{x}_s; s) - \right.$$
$$\left. \nabla_{\boldsymbol{x}_t} \log q_\phi(\boldsymbol{x}_s|\boldsymbol{x}_t; s, t)\right] - \frac{1}{2}\|\boldsymbol{v}_\psi(\boldsymbol{x}_s; s, t)\|_2^2\Big], \quad (10)$$
$$\min_\psi \; \mathbb{E}_{q_\phi(\boldsymbol{x}_s, \boldsymbol{x}_t; s, t)}\|\boldsymbol{f}_\psi(\boldsymbol{x}_s; s, t) - \log q_\phi(\boldsymbol{x}_s|\boldsymbol{x}_t; s, t)\|_2^2,$$

where $q_\phi(\boldsymbol{x}_s, \boldsymbol{x}_t; s, t) = q_{\text{trans}}(\boldsymbol{x}_s|\boldsymbol{x}_t; s, t)q_{\text{mix}}(\boldsymbol{x}_t; t)$.

**Shifting $f_\psi$ by Regularization**  The Nash-equilibrium of the two-stage optimization problem (10) is given by

$$\boldsymbol{f}_{\psi^*}(\boldsymbol{x}_s; s, t) = \nabla \log q_{\phi^*}(\boldsymbol{x}_s; s, t),$$
$$\phi^* \in \arg\min_\phi\{\mathbb{E}_{q_\phi(\boldsymbol{x}_s; s, t)}\|\boldsymbol{\delta}_\phi(\boldsymbol{x}_s; s, t)\|_2^2\}, \quad (11)$$
$$\boldsymbol{\delta}_\phi(\boldsymbol{x}_s; s, t) := \boldsymbol{S}_{\theta^*}(\boldsymbol{x}_s; s) - \nabla \log q_\phi(\boldsymbol{x}_s; s, t).$$

A natural initialization, therefore, would be to use the pre-trained score network $\boldsymbol{f}_{\psi_0}(\boldsymbol{x}_s; s, t) \approx \boldsymbol{S}_{\theta^*}(\boldsymbol{x}_s; s)$. During training, the optimal $\boldsymbol{f}_\psi$ is given by $\boldsymbol{f}_{\psi^*(\phi)}(\boldsymbol{x}_s; s, t) = \nabla \log q_\phi(\boldsymbol{x}_s; s, t)$. When $\nabla \log q_\phi(\boldsymbol{x}_s; s, t)$ deviates significantly from the target score model $\boldsymbol{S}_{\theta^*}(\boldsymbol{x}_s; s)$, this initialization strategy becomes inefficient for the second-stage optimization in (10). To address this mismatch, we follow Salimans et al. (2024) and adopt a regularization strategy

$$\min_\psi \; \mathbb{E}_{q_\phi(\boldsymbol{x}_s, \boldsymbol{x}_t; s, t)}\left[\|\boldsymbol{f}_\psi(\boldsymbol{x}_s; s, t) - \log q_\phi(\boldsymbol{x}_s|\boldsymbol{x}_t; s, t)\|_2^2\right.$$
$$\left. + \lambda\|\boldsymbol{f}_\psi(\boldsymbol{x}_s; s, t) - \boldsymbol{S}_{\theta^*}(\boldsymbol{x}_s; s)\|_2^2\right], \quad (12)$$

where $\lambda \geq 0$ is a hyperparameter controlling the strength of regularization. It can be shown that the optimal $f_{\tilde{\psi}^*(\phi)}$ of (12) is shifted towards $\boldsymbol{S}_{\theta^*}$ as follows

$$\boldsymbol{f}_{\tilde{\psi}^*(\phi)}(\boldsymbol{x}_s; s, t) = \frac{1}{1+\lambda}\nabla \log q_\phi(\boldsymbol{x}_s; s, t) + \frac{\lambda}{1+\lambda}\boldsymbol{S}_{\theta^*}(\boldsymbol{x}_s; s).$$

Although this additional regularization introduces bias into the second optimization in (10), we found that this bias in $\psi$ does not transfer to $\phi$, and the Nash equilibrium of $\phi$ remains consistent with (11). We provide a comprehensive statement in the theorem 3.1.

**Reformulation of Fused Loss**  In the fused training objective (6) of SiD, there is a mismatch in the two-stages training objective (10) when $\alpha$ is not $\frac{1}{2}$. Furthermore, when $\alpha > 1$, the fused loss function exhibits pathological behavior as it becomes negative when the inner optimization of $\boldsymbol{v}(\cdot; t)$ converges to its optimal solution

$$\hat{\mathcal{L}}_{\text{SiD}}(\phi) = \mathbb{E}_{\tilde{q}_\phi(\boldsymbol{x}_t; t)}(1 - \alpha)\|\nabla \log p(\boldsymbol{x}_s; s) - \nabla \log \tilde{q}_\phi(\boldsymbol{x}_t; t)\|_2^2,$$

However, within the framework of shifting $\boldsymbol{f}_\psi$, we can reformulate the SiD training objective to achieve unbiasedness while eliminating the aforementioned pathological behavior. Consider the scaled Fisher divergence $\mathrm{D}_\alpha(p(\cdot; s)\|q_\phi(\cdot; s, t))$ defined as

$$\mathrm{D}_\alpha(q_\phi(\cdot; s, t)\|p(\cdot; s)) := \mathbb{E}_{q_\phi(\boldsymbol{x}_s; t, s)}\frac{1}{4\alpha}\|\tilde{\boldsymbol{\delta}}_\phi(\boldsymbol{x}_s; s, t)\|_2^2,$$
$$\tilde{\boldsymbol{\delta}}_\phi(\boldsymbol{x}_s; s, t) := \nabla \log p(\boldsymbol{x}_s; s) - \nabla \log q_\phi(\boldsymbol{x}_s; s, t), \quad (13)$$

Then we can rewrite the above $\mathrm{D}_\alpha(p(\cdot; s)\|q_\phi(\cdot; s, t))$ as the maximum value of the optimization problem

$$\max_{\boldsymbol{v}_\psi} \; \mathbb{E}_{q_\phi(\boldsymbol{x}_s; s, t)}\Big[\boldsymbol{v}_\psi(\boldsymbol{x}_s; s, t)^T\left[\nabla \log p(\boldsymbol{x}_s; s) - \right.$$
$$\left. \nabla \log q_\phi(\boldsymbol{x}_s; s, t)\right] - \alpha\|\boldsymbol{v}_\psi(\boldsymbol{x}_s; s, t)\|_2^2\Big], \quad (14)$$

Similarly, we utilize (14) to reformulate the (13).

**Theorem 3.1.** *Optimization of $\mathrm{D}_\alpha(p(\cdot; s)\|q_\phi(\cdot; s, t))$ is equivalent to the two-stages optimization*

$$\min_\phi \; \mathbb{E}_{q_\phi(\boldsymbol{x}_s, \boldsymbol{x}_t; s, t)}\left[\boldsymbol{v}_\psi(\boldsymbol{x}_s; s, t)^T\left[\nabla \log p(\boldsymbol{x}_s; s) - \right.\right.$$
$$\left.\left. \nabla_{\boldsymbol{x}_s} \log q_\phi(\boldsymbol{x}_s|\boldsymbol{x}_t; s, t)\right] - \alpha\|\boldsymbol{v}_\psi(\boldsymbol{x}_s; s, t)\|_2^2\right], \quad (15)$$
$$\min_\psi \; \mathbb{E}_{q_\phi(\boldsymbol{x}_s, \boldsymbol{x}_t; s, t)}\left[\boldsymbol{v}_\psi(\boldsymbol{x}_s; s, t)^T\left[\nabla_{\boldsymbol{x}_s} \log q_\phi(\boldsymbol{x}_s|\boldsymbol{x}_t; s, t) - \right.\right.$$
$$\left.\left. \nabla \log p(\boldsymbol{x}_s; s)\right] + (1+\lambda)\alpha\|\boldsymbol{v}_\psi(\boldsymbol{x}_s; s, t)\|_2^2\right],$$

*where $\lambda \geq 0$ is a given regularization strength hyperparameter. The optimal $\boldsymbol{f}_{\psi^*(\phi)}(\boldsymbol{x}_s; s, t)$ of the second-stage optimization is given by*

$$\boldsymbol{f}_{\psi^*}(\phi)(\cdot; s, t) = \beta\nabla \log q_\phi(\cdot; s, t) + (1-\beta)\nabla \log p(\cdot; s), \quad (16)$$

*where $\beta = \frac{1}{2\alpha(1+\lambda)}$. The equilibrium of $\phi$ remains consistent with (11).*

It is straightforward to observe that (15) is the same as the SiD loss in (6). If $\nabla \log p(\boldsymbol{x}_s; s)$ is approximated by $\boldsymbol{S}_{\theta^*}(\boldsymbol{x}_s; s)$, the optimal $\boldsymbol{f}_{\tilde{\psi}^*}$ is given by

$$\boldsymbol{f}_{\tilde{\psi}^*}(\cdot; s, t) = \beta\nabla \log q_\phi(\cdot; s, t) + (1 - \beta)\nabla \log p(\cdot; s). \quad (17)$$

(17) indicates that when $\alpha > \frac{1}{2(1+\lambda)}$, then $\beta < 1$. The optimal $\boldsymbol{f}_{\tilde{\psi}^*(\phi)}(\boldsymbol{x}_s; s, t)$ in the second-stage optimization also shifts towards $\boldsymbol{S}_{\theta^*}(\boldsymbol{x}_s; s)$. This alignment facilitates learning when $\boldsymbol{f}_\psi$ is initialized as $\boldsymbol{S}_{\theta^*}$. The proof of Theorem 3.1 and its general version are shown in Appendix B.1. The training process of CoSIM is summarized in Algorithm 2 in Appendix A.1.

### 3.3. Parameterization of CoSIM

Consistency model (Song et al., 2023) is a family of generative models that learn a consistency function $\boldsymbol{G}_\phi(\boldsymbol{x}_t, t)$ to map the samples of distribution $p(\boldsymbol{x}_t; t)$ back to $p(\boldsymbol{x}_0; 0)$. Once the consistency function is well-trained, the consistency model can generate samples using multiple steps by iterating with $s < t$ as follows

$$\boldsymbol{x}_s = a(s)\boldsymbol{G}_\phi(\boldsymbol{x}_t, t) + \sigma(s)\boldsymbol{\epsilon}, \qquad (18)$$

which approximates $\boldsymbol{x}_0$ in (2) by $\boldsymbol{G}_\phi(\boldsymbol{x}_t, t)$, and $\boldsymbol{\epsilon} \sim \mathcal{N}(\boldsymbol{0}, \boldsymbol{I})$. Intuitively, this provides a parameterized form of the continuous transition kernel $q_{\text{trans}}(\boldsymbol{x}_s|\boldsymbol{x}_t; s, t)$. We hereafter adopt this setting for diffusion model distillation. One can expect that once $\boldsymbol{G}_\phi(\boldsymbol{x}_t, t)$ is perfectly trained with $\mathcal{L}_{\text{CSI-}f}(\phi)$, then $\boldsymbol{G}_\phi(\boldsymbol{x}_t, t)$ will also map the samples of distribution $p(\boldsymbol{x}_t; t)$ back to $p(\boldsymbol{x}_0; 0)$. We present this result in Proposition 3.2. See a detailed proof of Proposition 3.2 in Appendix B.2.

**Proposition 3.2** (Consistent Similarity). *Let the continuous transition kernel $q_{trans}(\boldsymbol{x}_s|\boldsymbol{x}_t; s, t)$ be parameterized as*

$$\boldsymbol{x}_s = [a(s)\boldsymbol{G}_\phi(\boldsymbol{x}_t, t) + \sigma(s)\boldsymbol{\epsilon}] \sim q_\phi(\boldsymbol{x}_s|\boldsymbol{x}_t; s, t),$$

*where $0 < s < t \leq T$, $\boldsymbol{\epsilon} \sim \mathcal{N}(\boldsymbol{0}, \boldsymbol{I})$, and $a(\cdot), \sigma(\cdot)$ are defined in (2). Then the optimal $\boldsymbol{G}_\phi(\cdot, t)$ obtained from the two-stage alternating optimization problem (15) also serves as a consistency function.*

With a well-trained continuous transition kernel $q_{\text{trans}}(\boldsymbol{x}_s|\boldsymbol{x}_t; s, t)$, we can iteratively sample from it over multiple steps to ultimately generate samples from $p(\boldsymbol{x}_0)$. To select the sequence of time points, we use the sampling scheme of EDM (Karras et al., 2022) as an initial choice and set a scale parameter for tuning with a greedy algorithm. We summarize the multistep sampling procedure in Algorithm 1.

**Error Bound of CoSIM** Building on the parameterized setting (18) of $q_{\text{trans}}(\boldsymbol{x}_s|\boldsymbol{x}_t; s, t)$, we denote the variational distribution family $\mathcal{Q} = \left\{ q_\phi | q_\phi(\boldsymbol{x}_s; s, t) = \int q_\phi(\boldsymbol{x}_s|\boldsymbol{x}_t; s, t) q_{\text{mix}}(\boldsymbol{x}_t) \mathrm{d}\boldsymbol{x}_t, \phi \in \Phi \right\}$, where $\Phi$ is the feasible domain of neural network parameters of $\boldsymbol{G}_\phi$ and $q_{\text{mix}}$ follows (8). Then we can define the optimal variational distribution in $\mathcal{Q}$ as

$$q_{\phi^*} \in \arg\min_{q_\phi \in \mathcal{Q}} \mathcal{L}_{\text{CSI}_\alpha}(q_\phi), \qquad (19)$$

---

**Algorithm 1** Inference of Continuous Semi-Implicit Models

**Input:** Continuous transition kernel $q_{\text{trans}}(\boldsymbol{x}_s|\boldsymbol{x}_t; s, t)$ and a sequence of time points $T = t_0 > t_1 > \cdots > t_k = 0$.
**Output:** The estimated samples of $p(\boldsymbol{x}_0)$.
Initialize $\boldsymbol{x}_0 \sim p(\boldsymbol{x}_T; T)$
**for** $n = 1$ **to** $k$ **do**
    Sample $\boldsymbol{x}_n \sim q_{\text{trans}}(\cdot|\boldsymbol{x}_{n-1}; t_n, t_{n-1})$
**end for**
**Output:** $\boldsymbol{x}_k$

---

where $\mathcal{L}_{\text{CSI}_\alpha[h]}$ is defined in (9) with the choice of criterion $\mathrm{D}_\alpha\left(p(\cdot; s)\|q_\phi(\cdot; s, t)\right)$ discussed in section 3.2.

We first introduce the following assumptions with a fixed $\lambda > 0$ and an early-stopping time $0 < \delta < T$.

As illustrated in Algorithm 2, the approximation errors arise from the inexact second-stage optimization of $\boldsymbol{f}_\psi$. We denote it as follows.

**Assumption 3.3** ($\boldsymbol{f}_\psi$ function error). The estimated auxiliary vector-valued function $\boldsymbol{f}_{\hat{\psi}(\phi)}(\boldsymbol{x}_t; t)$ in the two-stage optimization problem (15) is $\epsilon_{\text{f}}$-accurate, that is for all $s, t \in [\delta, T]$ with $s < t$ and $\phi \in \Phi$:

$$\mathbb{E}_{q_\phi(\boldsymbol{x}_s; s, t)} \|\boldsymbol{f}_{\hat{\psi}(\phi)}(\boldsymbol{x}_s; s, t) - \boldsymbol{f}_{\psi^*(\phi)}(\boldsymbol{x}_s; s, t)\|_2^2 \leq \epsilon_{\text{f}}^2. \quad (20)$$

Let the $q_{\hat{\phi}}(\boldsymbol{x}_s; s, t)$ be the one obtained by the first stage optimization problem with the $\epsilon_{\text{f}}$-accurate $\boldsymbol{f}_{\hat{\psi}(\phi)}$

$$q_{\hat{\phi}}(\cdot; s, t) \in \arg\min_{q_\phi \in \mathcal{Q}} \mathbb{E}_{q_\phi(\boldsymbol{x}_s, \boldsymbol{x}_t; s, t)} \Big[ \boldsymbol{u}_{\hat{\psi}(\phi)}(\boldsymbol{x}_s; s, t)^T [\boldsymbol{S}_{\theta^*}(\boldsymbol{x}_s; s)$$
$$- \nabla_{\boldsymbol{x}_s} \log q_\phi(\boldsymbol{x}_s|\boldsymbol{x}_t; s, t)] - \alpha \|\boldsymbol{u}_{\hat{\psi}(\phi)}(\boldsymbol{x}_s; s, t)\|_2^2 \Big],$$

where $\boldsymbol{u}_{\hat{\psi}(\phi)}(\boldsymbol{x}_s; s, t) := \log p(\boldsymbol{x}_s; s) - \boldsymbol{f}_{\hat{\psi}(\phi)}(\boldsymbol{x}_s, \boldsymbol{x}_t; s, t)$.

Then we are ready to state the following result.

**Proposition 3.4** (Error bound of one-step map, Yu & Zhang (2023)). *Suppose the assumptions [3.3] hold. Then for any $0 < s < t < T$, the Fisher divergence between the target distribution $p(\boldsymbol{x}_\delta)$ and the estimated distribution of CoSIM is bounded as follows*

$$FI(q_{\hat{\phi}}(\cdot; s, t)\|p(\cdot; s)) \lesssim FI(q_{\phi^*(\boldsymbol{x}_s; s, t)}\|p(\cdot, s)) + \varepsilon_f^2.$$

The above results show that the approximation error of inexact $\boldsymbol{f}_\psi$ only attribute an extra term $\varepsilon_{\text{f}}^2$ to the approximation accuracy of CoSIM. As long as the approximation error $\varepsilon_{\text{f}}$ is small, the two stages optimization of CoSIM with regularization strength $\lambda > 0$ will provide the same approximation accuracy as the optimization with the original training objective (13).

## 3.4. Insights on the Benefits of Multi-Step Methods

As discussed in section 3.2, we adopt the same type of SIVI objective ($\mathcal{L}_{\text{SIVI-}f}$) as in SiD (Zhou et al., 2024b) for diffusion model distillation. However, unlike the deterministic one-step generation scheme of SiD, continuous semi-implicit models (CoSIM) allow us to generate samples from $p(\boldsymbol{x}_0)$ through multiple steps. A notable advantage of CoSIM is that it injects randomness at each generation step, producing more diverse samples—an observation consistent with the benefits of stochastic diffusion models (Song et al., 2020). Moreover, as we show next, CoSIM can generate samples closer to the target distribution than the one-step distillation model by employing multistep sampling.

Since the optimal variational distribution $q_{\phi^*}(\boldsymbol{x}_s; s, t)$ is defined on $\mathcal{Q}$, this family itself will introduce approximation gaps due to the approximation capacity of $\boldsymbol{G}_\phi(\boldsymbol{x}_t, t)$. Since $\boldsymbol{G}_\phi(\boldsymbol{x}_t, t)$ maps the distribution $p(\boldsymbol{x}_t; t)$ back to $p(\boldsymbol{x}_0; 0)$, the approximation error is expected to grow with $t$. Following the assumption in previous work Chen et al. (2023), we scale the error term of $\boldsymbol{G}_\phi(\boldsymbol{x}_t, t)$ accordingly.

**Assumption 3.5** (Consistency function error). For all $t \in [\delta, T]$, the approximation gaps between $\nabla \log q_{\phi^*}(\boldsymbol{x}_\delta, \delta, t)$ and $\nabla \log p(\boldsymbol{x}_\delta; \delta)$ is bounded in $L^2(q_{\phi^*})$:

$$\mathbb{E}_{q_{\phi^*}(\boldsymbol{x}_\delta; \delta, t)} \|\nabla \log p(\boldsymbol{x}_\delta; \delta) - \nabla \log q_{\phi^*}(\boldsymbol{x}_\delta; \delta, t)\|_2^2 \leq \varepsilon_{\text{g}}(t)^2,$$

where the error term is scaled by $\varepsilon_{\text{g}}(t)^2 := 2\varepsilon_{\text{c}}^2 \text{FI}(p(\cdot; t + \delta) \| p(\cdot; \delta))$, and error $\varepsilon_{\text{c}}$ characterizes the approximation capacity of the consistency function family $\{\boldsymbol{G}_\phi | \phi \in \Phi\}$. $\text{FI}(\cdot \| \cdot)$ denotes the Fisher divergence.

We note that the divergence $\text{FI}(p(\cdot; t + \delta) \| p(\cdot, \delta))$ grows as $t$ increases to $T$. Furthermore, when $\{\boldsymbol{G}_\phi \mid \phi \in \Phi\}$ contains only the identity mapping, the aforementioned upper bound holds with $\varepsilon_{\text{c}}^2 = 1$, validating the reasonableness of assumption 3.5. More details can be found in the proof of Proposition 3.4 in Appendix B.2.

**Assumption 3.6.** The perturbed distribution $p(\boldsymbol{x}_\delta; \delta)$ satisfies a logarithmic Sobolev inequality (LSI) as follows.

$$\text{ent}(f^2) \leq 2L_{\text{LSI}} \mathbb{E}_{p(\boldsymbol{x}_\delta; \delta)} \Gamma(f), \quad \forall f \in \mathcal{C}^\infty(\mathbb{R}^d), \quad (21)$$

where $\text{ent}(f^2) = \mathbb{E}_{p(\boldsymbol{x}_\delta; \delta)}(f^2(\log f^2 - \mathbb{E}_{p(\boldsymbol{x}_\delta; \delta)} f^2))$ is the entropy of $f^2$, $\Gamma(f) = \|\nabla f\|_2^2$ and $L_{\text{LSI}}$ is the LSI constant.

The LSI assumption is a standard assumption in the analysis of diffusion models (Lee et al., 2023), and previous works show that the LSI inequality holds when $p(\boldsymbol{x}_0; 0)$ is a bounded distribution (Chen et al., 2021). More details can be found in the proof of proposition 3.4.

Then we investigate the smoothness assumption of the consistency function $\boldsymbol{G}_\phi(\boldsymbol{x}_t, t)$. Following the preconditioning strategy of neural network proposed in (Karras et al., 2022;

Song et al., 2023), we employ the consistency function as $\boldsymbol{G}_\phi(\boldsymbol{x}_t, t)$ as follows

$$\boldsymbol{G}_\phi(\boldsymbol{x}_t, t) = c_{\text{skip}}(t)\boldsymbol{x}_t + c_{\text{out}}(t)\boldsymbol{F}_\phi(c_{\text{in}}(t)\boldsymbol{x}_t, t), \quad (22)$$

where $c_{\text{skip}}$ modulates the skip connection and $c_{\text{in}}, c_{\text{out}}$ are the scaling factors of network input and output.

**Assumption 3.7** (Smoothness of neural network). The neural network $\boldsymbol{F}_\phi(\boldsymbol{x}, t)$ in (22) is $L_f$-Lipschitz continuous on the variable $\boldsymbol{x}$ with constant $L_f > 1$ for all $t \in [\delta, T]$.

The Lipschitz continuous assumption is naturally used in previous works (Song et al., 2023; Lyu et al., 2024; Li et al., 2024). However, we adopt a practical preconditioning and impose the Lipschitz condition on $\boldsymbol{F}_\phi$ rather than $\boldsymbol{G}_\phi$. Moreover, these two Lipschitz conditions are equivalent under the variance preserving (VP) scheme, which is shown in Appendix B.4.

Based on the above error bound in proposition 3.4, we can give a convergence bound for the multistep sampling of CoSIM similar to Lyu et al. (2024).

**Proposition 3.8** (Multistep Wasserstein Distance Error). *Under Assumptions [3.3-3.7], consider a sequence of time points $T = t_0 \geq t_1 = \cdots = t_{K-1} = t_{mid} > t_K = 0$. Let $q_{\hat{\phi}}^{(K)}(\cdot)$ denote the $K$-step estimated distribution by CoSIM. Then for variance preserving scheme of (1), there exists $t_{mid} = \mathcal{O}(\log L_f)$ for variance preserving scheme of (1) such that*

$$W_2(q_{\hat{\phi}}^{(K)}, p(\cdot; 0)) \lesssim \delta^2 d + \left(\frac{3}{4}\right)^{K-1} \mathcal{E}_{W_2^2}^{\frac{1}{2}}(T) + \mathcal{E}_{W_2^2}^{\frac{1}{2}}(t_{mid}),$$

*where $\mathcal{E}_{W_2^2}(t)$ denotes the Wasserstein distance bound between the one-step CoSIM estimate $q_{\hat{\phi}}(\cdot; \delta; t)$ and $p(\cdot; \delta)$. This bound is controlled by the error terms from Assumptions [3.3-3.7], with the explicit formulation detailed in Appendix B.4. For variance exploding scheme of (1), the above results hold for some $t_{mid} = \mathcal{O}(L_f)$.*

Intuitively, since the Wasserstein distance bound $\mathcal{E}_{W_2^2}(t)$ is increasing with $t$, the benefit of multistep sampling lies in reducing the error bound from $\mathcal{E}_{W_2^2}^{\frac{1}{2}}(T)$ of the one-step model to a smaller one $\mathcal{E}_{W_2^2}^{\frac{1}{2}}(t_{\text{mid}})$. We provide a detailed proof of Proposition 3.8 in Appendix B.4.

## 4. Experiments

In this section, we explore the performance of CoSIM on the unconditional image generation task on CIFAR-10 ($32 \times 32$) and the conditional image generation task on ImageNet ($64 \times 64$) and ImageNet ($512 \times 512$). For all experiments, the network architecture of the transition kernel $q_\phi(\cdot | \boldsymbol{x}_t; s, t)$ and the auxiliary function $\boldsymbol{v}_\psi(\boldsymbol{x}_s; s, t)$ is almost identical to the pre-trained score network $\boldsymbol{S}_{\theta^*}(\boldsymbol{x}_s; s)$ (Karras et al.,

2022), except for an additional time embedding network of $t$. For the purpose of minimal changes, this is done by duplicating the time embedding network of $s$ in the score network architecture and summing up the time embedding of $s$ and $t$ together, leading to only a $4\%$ increase in parameters. The implementation of CoSIM is available at `https://github.com/longinYu/CoSIM`.

We use the well-established metric Frechet Inception Distance (Salimans et al., 2016) (FID) to measure the quality of the generated images. Furthermore, as the FID score often unfairly favors the models trained with GAN losses and penalizes the diffusion models, we consider an additional metric - FD-DINOv2 (Stein et al., 2024), which replaces the InceptionV3 (Szegedy et al., 2016) encoder of FID by DINOv2 (Oquab et al., 2024) to better align with human perception. Both FID and FD-DINOv2 are evaluated on 50K generated images and the whole training set, which means 50K images from CIFAR10 (Krizhevsky & Hinton, 2009) training split and 1,281,167 images from ImageNet (Deng et al., 2009).

The training setup for CoSIM is basically kept the same as the underlying pre-trained score models, only with a different optimization objective in Section 3.2. In practice, we adopt the setting $\alpha = 1.2$ as in Zhou et al. (2024b). In contrast to the SiD loss, which uses $\alpha(1 + \lambda) = 0.5$, we choose to increase the regularization strength $\lambda$ such that $\alpha(1 + \lambda) = 1$ for simplicity. The time schedule is detailed in Appendix A.1. We re-use most of the optimizer settings from the pretrained score models with some slight tweakings such as learning rate and batch size, more details can be found in Appendix C.

### 4.1. Unconditional Image Generation

**CIFAR-10** $(32 \times 32)$  On the unconditional image generation task on CIFAR-10, the pre-trained score network $\boldsymbol{S}_{\theta^*}(\boldsymbol{x}_s; s)$ is from Karras et al. (2022).

Table 1 reports the generation quality measured by FID and FD-DINOv2. We see that our CoSIM reaches the same FID as the teacher (VP-EDM) with only 4 NFE. Although the 2-step CoSIM reaches an FID above 2, this is still considerably lower than many of the leading methods. The reasons for CoSIM's lagging behind other methods can be attributed to our model size, which is significantly smaller than others. On the FD-DINOv2, our CoSIM achieves state-of-the-art results compared to both trained-from-scratch and distilled models. Specifially, CoSIM attains an FD-DINOv2 of 113.51 with 4 NFE, outperforming the best-ever PFGM++ (Xu et al., 2023, an FD-DINOv2 of 141.65) by a large margin. We also note that this leading status of CoSIM still holds with 2 NFE. Furthermore, the FID metric's inclination towards GAN-based models (Stein et al., 2024) may partly explain why CTM achieves the best FID score, rather than

*Table 1.* Unconditional generation quality on CIFAR-10 ($32 \times 32$). Results with asterisks (*) are tested by ourselves with the official codes and checkpoints. ECT does not provide official checkpoints, so its FD-DINOv2 score ($^\dagger$) comes from our re-implementation. The other results are from the original papers. The best result is marked in **black bold font** and the second best result is marked in **brown bold font**.

| Method | #Params | NFE | FID ($\downarrow$) | FD-DINOv2 ($\downarrow$) |
|---|---|---|---|---|
| CIFAR:Test Split | - | - | 3.15 | 31.07 |
| DDPM (Ho et al., 2020) | | 1000 | 3.17 | - |
| DDIM (Song et al., 2021) | | 100 | 4.16 | - |
| TDPM+ (Zheng et al., 2023a) | | 100 | 2.83 | - |
| DPM-Solver3 (Lu et al., 2022) | | 48 | 2.65 | - |
| VP-EDM (Karras et al., 2022) | 56M | 35 | 1.97 | 168.01* |
| VP-EDM+LEGO-PR (Zheng et al., 2023b) | | 35 | 1.88 | - |
| PFGM++ (Xu et al., 2023) | | 35 | 1.91 | 141.65 |
| HSIVI-SM (Yu et al., 2023) | | 15 | 4.17 | - |
| CD-LPIPS (Song et al., 2023) | | 1 | 3.55 | - |
| iCT (Song & Dhariwal, 2024) | | 2 | 2.93 | - |
| | | 1 | 2.83 | - |
| | | 2 | 2.46 | - |
| sCT (Lu & Song, 2024) | | 1 | 2.97 | - |
| | | 2 | 2.06 | - |
| CTM (Kim et al., 2023) | 324M | 1 | 1.98 | 237.08* |
| CTM (Kim et al., 2023) | 324M | 2 | 1.87 | 210.64* |
| CTM (Kim et al., 2023) | 324M | 4 | **1.84*** | 197.59* |
| ECT (Geng et al., 2024) | 56M | 2 | 2.11 | 211.17$^\dagger$ |
| SiD (Zhou et al., 2024b) | 56M | 1 | 1.92 | 148.17* |
| **CoSIM (ours)** | 56M | 2 | 2.40 | 116.92 |
| **CoSIM (ours)** | 56M | 4 | 1.97 | 113.51 |
| CoSIM(ours) | 56M | 6 | 1.96 | **111.98** |

FD-DINOv2. Finally, we observe a consistent decrease in both FID and FD-DINOv2 as the NFE increases from 2 to 4, which accords with our theoretical analysis in Section 3.4.

### 4.2. Conditional Image Generation

**ImageNet** $(64 \times 64)$  Similarly to Section 4.1, the network structure and the checkpoint of the pre-trained score model $\boldsymbol{S}_{\theta^*}(\boldsymbol{x}_s; s)$ are from Karras et al. (2022).

We report the conditional generation quality of different models in Table 3. We see that the VP-EDM (the teacher model) can get a worse FID with 999 NFE, while the FD-DINOv2 decreases monotonically as NFE increases. This consistency in FD-DINOv2 further verifies the reasonability of introducing FD-DINOv2 for measuring the generation quality of diffusion-based models. After distillation, our CoSIM produces an FID of 1.46 with 4 NFE, reaching a similar performance to VP-EDM with 999 NFE and outperforming most methods in Table 3. Our CoSIM also achieves state-of-the-art results on FD-DINOv2, setting the lowest record at 56.66 with only 4 NFE. We also notice that Moment Matching (Salimans et al., 2024) reaches a better FID score compared to our CoSIM, which we attribute to the smaller network architecture employed by us.

**ImageNet** $(512 \times 512)$  FD-DINOv2 is a new metric proposed very recently (Stein et al., 2024). Most of the models published before FD-DINOv2 naturally did not consider metric in their experiments and thus may not set up their

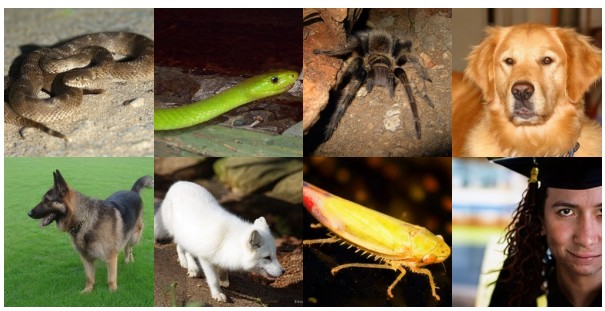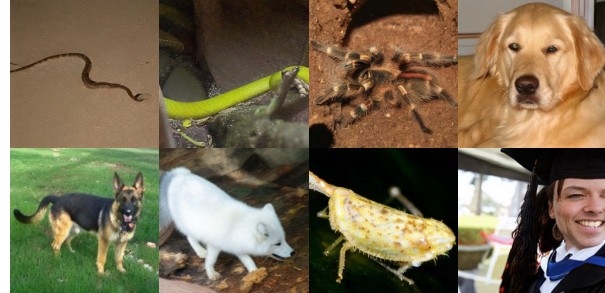

*Figure 2.* Conditionally generated images on ImageNet ($512 \times 512$). The two batches of images are generated from 4-step (**left**) and 2-step (**right**) CoSIM L model respectively, with identical initial noise and class labels.

*Table 2.* Conditional generation quality measured by FD-DINOv2 on ImageNet ($512 \times 512$). Results of SiD are reported from Zhou et al. (2025).

| Method | #Params | NFE | FD-DINOv2 ($\downarrow$) |
|---|---|---|---|
| EDM2-S-DINOv2 (Karras et al., 2024) | 280M | 63 | 68.64 |
| EDM2-M-DINOv2 (Karras et al., 2024) | 498M | 63 | 58.44 |
| EDM2-L-DINOv2 (Karras et al., 2024) | 778M | 63 | 52.25 |
| EDM2-XL-DINOv2 (Karras et al., 2024) | 1.1B | 63 | 45.96 |
| EDM2-XXL-DINOv2 (Karras et al., 2024) | 1.5B | 63 | 42.84 |
| SiD-S (Zhou et al., 2024b) | 280M | 1 | **65.08** |
| SiD-M (Zhou et al., 2024b) | 498M | 1 | 55.92 |
| SiD-L (Zhou et al., 2024b) | 777M | 1 | 56.25 |
| **CoSIM-S (ours)** | 280M | 2 | 67.81 |
| | 280M | 4 | 67.71 |
| **CoSIM-M (ours)** | 498M | 2 | 53.35 |
| | 498M | 4 | **51.57** |
| **CoSIM-L (ours)** | 778M | 2 | 46.41 |
| | 778M | 4 | **41.79** |

*Table 3.* Conditional generation quality on ImageNet ($64 \times 64$). Results with asterisks ($^*$) are tested by ourselves with the official codes and checkpoints. The other results are from the original papers. The best result is marked in **black bold font** and the second best result is marked in **brown bold font**.

| Method | #Params | NFE | FID ($\downarrow$) | FD-DINOv2 ($\downarrow$) |
|---|---|---|---|---|
| DDPM (Ho et al., 2020) | | 250 | 11.00 | - |
| TDPM+ (Zheng et al., 2023a) | | 1000 | 1.62 | - |
| DPM-Solver3 (Lu et al., 2022) | | 50 | 17.52 | - |
| VP-EDM (Karras et al., 2022) | 296M | 79 | 2.64 | 107.07* |
| | 296M | 511 | 1.36 | 79.82* |
| | 296M | 999 | 1.41* | 72.67* |
| VP-EDM+LEGO-PR (Zheng et al., 2023b) | | 250 | 2.16 | - |
| HSIVI-SM (Yu et al., 2023) | | 15 | 15.49 | - |
| CD-LPIPS (Song et al., 2023) | | 1 | 6.20 | - |
| | | 2 | 4.70 | - |
| iCT (Song & Dhariwal, 2024) | | 1 | 4.02 | - |
| | | 2 | 3.20 | - |
| sCT (Lu & Song, 2024) | | 1 | 2.04 | - |
| | | 2 | 1.48 | - |
| CTM (Kim et al., 2023) | 604M | 1 | 1.92 | 163.47* |
| | 604M | 2 | 1.73 | 159.04* |
| Moment Matching (Salimans et al., 2024) | 400M | 1 | 3.0 | - |
| | 400M | 2 | 3.86 | - |
| | 400M | 4 | 1.50 | - |
| | 400M | 8 | **1.24** | - |
| ECT (Geng et al., 2024) | 296M | 2 | 1.67 | $\sim 150$ |
| Diff-Instruct (Luo et al., 2023) | 296M | 1 | 5.57 | - |
| SiD (Zhou et al., 2024b) | 296M | 1 | 1.52 | **79.15***|
| **CoSIM (ours)** | 296M | 2 | 3.35 | 108.99 |
| **CoSIM (ours)** | 296M | 4 | **1.46** | **58.66** |

models perfectly for getting a good FD-DINOv2 result. To make a more convincing comparison, we further test CoSIM on ImageNet ($512 \times 512$) against EDM2 (Karras et al., 2024) which also incorporates FD-DINOv2 benchmark in their original paper. This experiment also validates the scalability of our approach.

We test our method on three different model sizes (S, M, L) from Karras et al. (2024), which are significantly larger than those models tested in CIFAR $32 \times 32$ and Imagenet $64 \times 64$. Also, with a steady increase in parameter numbers from S to M and L, we showcase the scalability of our approach by generating superior results on larger models.

Tables [2-4] report the conditional generation quality measured by FD-DINOv2 and FID of different models. As the model size grows, the generation quality of both models improves consistently. For all the model sizes (S,M,L), our CoSIM with only 2 NFE surpasses the teacher with 63 NFE, demonstrating the effective distillation ability of CoSIM for handling high-dimensional samples and larger models. Due to the GPU memory limit, we do not test CoSIM on XL and XXL models but expect its similarly superior performance, since CoSIM-L with 4 NFE already beats EDM2-XXL-dino.

We also provide the conditionally generated images from CoSIM in Figure 2. We ensure that the 4-step generation and 2-step generation start from the same initial noise and class labels, and show that there is a clearly observable difference. Specifically, 2-step samples are good enough to capture the majority of contents, but 4-step samples tend to capture more fine-grained details, e.g., the human eyes, cricket legs, and snake bodies shown in this figure.

## 5. Conclusion

We presented CoSIM, a continuous-time semi-implicit framework for accelerating diffusion models through stochastic multi-step generation. Unlike the discrete-time sequential training in hierarchical semi-implicit variational inference, CoSIM leverages a continuous-time framework

Table 4. Conditional generation quality measured by FID on ImageNet ($512 \times 512$).

| Method | #Params | NFE | FID ($\downarrow$) |
|---|---|---|---|
| EDM2-S-FID (Karras et al., 2024) | 280M | 63 | 2.56 |
| EDM2-M-FID (Karras et al., 2024) | 498M | 63 | 2.25 |
| EDM2-L-FID (Karras et al., 2024) | 778M | 63 | 2.06 |
| EDM2-XL-FID (Karras et al., 2024) | 1.1B | 63 | 1.96 |
| EDM2-XXL-FID (Karras et al., 2024) | 1.5B | 63 | 1.91 |
| sCD-S (Lu & Song, 2024) | 280M | 2 | **2.50** |
| sCD-M (Lu & Song, 2024) | 498M | 2 | 2.26 |
| sCD-L (Lu & Song, 2024) | 778M | 2 | 2.04 |
| SiD-S (Zhou et al., 2024b) | 280M | 1 | 2.71 |
| SiD-M (Zhou et al., 2024b) | 498M | 1 | 2.06 |
| SiD-L (Zhou et al., 2024b) | 778M | 1 | 1.91 |
| **CoSIM-S (ours)** | 280M | 2 | 2.66 |
|  | 280M | 4 | 2.56 |
| **CoSIM-M (ours)** | 498M | 2 | 1.95 |
|  | 498M | 4 | **1.93** |
| **CoSIM-L (ours)** | 778M | 2 | 1.84 |
|  | 778M | 4 | **1.83** |

to approximate continuous transition kernels. By introducing a framework of equilibrium point shifting, we establish theoretical guarantees for unbiased two-stage optimization while resolving the pathological behavior inherent in SiD training objectives. Through careful parameterization of continuous transition kernels, we provide a novel method for multistep distillation of generative models training on a distributional level. In experiments, we demonstrate that CoSIM achieves comparable or superior FID while only requiring fewer training iterations and utilizing similar or smaller neural networks compared to existing one-step distillation methods and consistency model variants. Furthermore, CoSIM achieves state-of-the-art performance on both unconditional and conditional image generation tasks as measured by FD-DINOv2.

**Limitations** For diffusion distillation, CoSIM training involves three models: the generator $G_\theta$, the auxiliary function $f_\psi$ and the target score model $S_{\theta^*}$. As a result, CoSIM incurs high memory consumption due to the involvement of multiple models. Additionally, since CoSIM initializes the generator $G_\phi$ and the function $f_\psi$ from a pre-trained target score model, it limits the flexibility of $G_\phi$ to leverage larger architectures for modeling the continuous transition kernels. Consequently, the one-step generation quality of CoSIM is lower than that of modern one-step distillation models such as SiD and sCD, we defer this aspect to future research.

## Impact Statement

This paper presents work whose goal is to advance the field of Machine Learning. There are many potential societal consequences of our work, none which we feel must be specifically highlighted here.

## Acknowledgements

This work was supported by National Natural Science Foundation of China (grant no. 12201014, grant no. 12292980 and grant no. 12292983). The research of Cheng Zhang was support in part by National Engineering Laboratory for Big Data Analysis and Applications, the Key Laboratory of Mathematics and Its Applications (LMAM) and the Key Laboratory of Mathematical Economics and Quantitative Finance (LMEQF) of Peking University. The research of S.-H. Gary Chan was supported, in part, by Research Grants Council Collaborative Research Fund (under grant number C1045-23G). The authors appreciate the anonymous ICML reviewers for their constructive feedback.

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

---

**Algorithm 2** Training procedure of Continuous Semi-Implicit Models (CoSIM)

---

**Input:** Dataset $\mathcal{D}$, pretrained score model $\boldsymbol{S}_{\theta^*}(\boldsymbol{x}_s; s)$, continuous transition kernel $q_\phi(\boldsymbol{x}_s|\boldsymbol{x}_t; s, t)$, auxiliary vector-valued function $\boldsymbol{f}_\psi(\boldsymbol{x}_s; s, t)$ and $\lambda, \alpha$ s.t. $\alpha = 1.2$, $\alpha(1 + \lambda) = 1.0$.
**Output:** Continuous transition kernel $q_\phi(\boldsymbol{x}_s|\boldsymbol{x}_t; s, t)$.
Initialization: $\phi \leftarrow \theta^*, \psi \leftarrow \theta^*$ with require_all=False and iteration number $n = 0$.
**repeat**
    Sample continuous time points $s \sim \pi(s)$ and $t \sim \pi(t|s)$.
    Sample $\boldsymbol{x}_0 \sim \mathcal{D}$ and $\boldsymbol{x}_t \sim p(\boldsymbol{x}_t|\boldsymbol{x}_0; t, 0)$.
    Sample $\boldsymbol{x}_s \sim q_\phi(\boldsymbol{x}_s|\boldsymbol{x}_t; s, t)$.
    $\boldsymbol{u}_\psi(\boldsymbol{x}_s; s, t) \to \boldsymbol{S}_{\theta^*}(\boldsymbol{x}_s; s) - \boldsymbol{f}_\psi(\boldsymbol{x}_s; s, t)$.
    **if** $n$ is odd **then**
        Update $\phi \leftarrow \phi - \eta w_1(s) \nabla_\phi \Big\{ \boldsymbol{u}_\psi(\boldsymbol{x}_s; s, t)^T [\boldsymbol{S}_{\theta^*}(\boldsymbol{x}_s; s) - \nabla_{\boldsymbol{x}_s} \log q_\phi(\boldsymbol{x}_s|\boldsymbol{x}_t; s, t)] - \alpha \|\boldsymbol{u}_\psi(\boldsymbol{x}_s; s, t)\|_2^2 \Big\}$.
    **else**
        Update $\psi \leftarrow \psi - \eta w_2(s) \nabla_\psi \Big\{ \boldsymbol{u}_\psi(\boldsymbol{x}_s; s, t)^T [\nabla_{\boldsymbol{x}_s} \log q_\phi(\boldsymbol{x}_s|\boldsymbol{x}_t; s, t) - \boldsymbol{S}_{\theta^*}(\boldsymbol{x}_s; s)] + \alpha(1 + \lambda) \|\boldsymbol{u}_\psi(\boldsymbol{x}_s; s, t)\|_2^2 \Big\}$.
    **end if**
    $n \leftarrow n + 1$.
**until** convergence

---

## A. SIVI Objectives

### A.1. Details of training procedure

**Choice of Time Schedule** We now discuss the design principles of the time schedule $\pi(s, t)$ defined over $[\delta, T]$. First, we decompose $\pi(s, t)$ into the marginal distribution $\pi(s)$ and the conditional distribution $\pi(t|s)$. For $\pi(s)$, we adopt the EDM time schedule (Karras et al., 2022):

$$s = (\sigma_{\min}^{\frac{1}{\rho}} + r_s \cdot (\sigma_{\max}^{\frac{1}{\rho}} - \sigma_{\min}^{\frac{1}{\rho}}))^\rho, \quad r_s \sim \text{Uniform}[0, 1],$$

where $\sigma_{\min} = \delta$, $\sigma_{\max} = T$, and $\rho = 7.0$. The conditional distribution $\pi(t|s)$ is parameterized as:

$$t = (\sigma_{\min}^{\frac{1}{\rho}} + r_t \cdot (\sigma_{\max}^{\frac{1}{\rho}} - \sigma_{\min}^{\frac{1}{\rho}}))^\rho, \quad r_t = \min\{r_s + \gamma, 1\},$$

where $\gamma \sim \pi(\gamma|r_s)$. Since the continuous transition kernel $q_{\text{trans}}(\boldsymbol{x}_s|\boldsymbol{x}_t; s, t)$ learns the mapping from distribution $p(\cdot; t)$ to $p(\cdot; s)$, the learning complexity increases with the time difference between $t$ and $s$. Therefore, we design the distribution $\pi(\gamma|r_s)$ to assign higher probabilities to larger time difference $\gamma$.

$$\gamma = B \cdot \text{Uniform}[0, 1] \cdot \text{Uniform}\{1, \frac{1}{2}, \cdots, \frac{1}{R}\} + (1 - B), \quad B \sim \text{Bernoulli}(\frac{1}{2}),$$

where $R > 1$ is a hyperparameter controlling the probability of sampling smaller values of $\gamma$.

Leveraging the aforementioned time schedule and adopting the weight functions $w_i(s)$ ($i = 1, 2$) from SiD (Zhou et al., 2024b), we present our complete training procedure in Algorithm 2.

## B. Theoretical Result

### B.1. Proof of Theorem 3.1

Here we provide a general version of theorem 3.1 and illustrate the scaled Fisher divergence within the framework of generalized Fisher divergence (Cheng et al., 2023). Consider the generalized Fisher divergence equipped with a strictly smooth convex function $h(\boldsymbol{x})$

$$\text{GFI}[h](p(\cdot; s) \| q_\phi(\cdot; s, t)) := \mathbb{E}_{q_\phi(\boldsymbol{x}_s; t, s)} h\left(\nabla \log p(\boldsymbol{x}_s; s) - \nabla \log q_\phi(\boldsymbol{x}_s; s, t)\right),$$

where $h(\boldsymbol{0}) = 0$ and $h(\boldsymbol{x}) > 0$ for $\boldsymbol{x} \neq \boldsymbol{0}$.

**Theorem B.1.** *Let the variational distribution $q_\phi(\boldsymbol{x}_s; s, t)$ defined as in (7) and $\boldsymbol{v}_\psi(\boldsymbol{x}_s; s, t) := \nabla \log p(\boldsymbol{x}_s; s) - \boldsymbol{f}_\psi(\boldsymbol{x}_s; s, t)$. Then the optimization of GFI[h] $(p(\cdot; s) \| q_\phi(\cdot; s, t))$ is equivalent to the two-stages alternating optimization problem as follows*

$$\min_\phi \quad \mathbb{E}_{q_\phi(\boldsymbol{x}_s, \boldsymbol{x}_t; s, t)} \left[ \boldsymbol{v}_\psi(\boldsymbol{x}_s; s, t)^T \left[ \nabla \log p(\boldsymbol{x}_s; s) - \nabla_{\boldsymbol{x}_s} \log q_\phi(\boldsymbol{x}_s | \boldsymbol{x}_t; s, t) \right] - h^*(\boldsymbol{v}_\psi(\boldsymbol{x}_s; s, t)) \right], \tag{23}$$

$$\min_\psi \quad \mathbb{E}_{q_\phi(\boldsymbol{x}_s, \boldsymbol{x}_t; s, t)} \left[ \boldsymbol{v}_\psi(\boldsymbol{x}_s; s, t)^T \left[ \nabla_{\boldsymbol{x}_s} \log q_\phi(\boldsymbol{x}_s | \boldsymbol{x}_t; s, t) - \nabla \log p(\boldsymbol{x}_s; s) \right] + (1 + \lambda) h^*(\boldsymbol{v}_\psi(\boldsymbol{x}_s; s, t)) \right],$$

*where $\lambda \geq 0$ is a given regularization strength hyperparameter and $h^*(\cdot)$ is the Legendre transformation of $h(\cdot)$. Specifically, if $h(\boldsymbol{x}) = \frac{1}{4\alpha} \|\boldsymbol{x}\|_2^2$ and $\nabla \log p(\boldsymbol{x}_s; s)$ is approximated by $\boldsymbol{S}_{\theta^*}(\boldsymbol{x}_s; s)$, (23) becomes the SiD loss (6). The optimal $\boldsymbol{f}_{\tilde{\psi}^*(\phi)}(\boldsymbol{x}_s; s, t)$ of the second-stage optimization is given by*

$$\boldsymbol{f}_{\tilde{\psi}^*(\phi)}(\cdot; s, t) = \beta \nabla \log q_\phi(\cdot; s, t) + (1 - \beta) \boldsymbol{S}_{\theta^*}(\cdot; s), \tag{24}$$

*where $\beta = \frac{1}{2\alpha(1+\lambda)}$. Similarly, the equilibrium of $\phi$ remains consistent with (11).*

*Proof of Theorem B.1.* We can write the optimization problem with the generalized Fisher divergence GFI[h] as

$$\min_{q_\phi} \quad \mathbb{E}_{q_\phi(\boldsymbol{x}_s; t, s)} h(\nabla \log p(\boldsymbol{x}_s; s) - \nabla \log q_\phi(\boldsymbol{x}_s; s, t)). \tag{25}$$

Since the Legendre transformation of $h(\boldsymbol{x})$ is that $h^*(\boldsymbol{y}) = \max_{\boldsymbol{x}} \{\boldsymbol{y}^T \boldsymbol{x} - h(\boldsymbol{x})\}$ and $h^{**}(\cdot) = h(\cdot)$, we can rewrite the optimization problem as

$$\min_{q_\phi} \max_{\boldsymbol{v}_\psi(\cdot; s, t)} \quad \mathbb{E}_{q_\phi(\boldsymbol{x}_s; s, t)} \left[ \boldsymbol{v}_\psi(\boldsymbol{x}_s; s, t)^T \left[ \nabla \log p(\boldsymbol{x}_s; s) - \nabla \log q_\phi(\boldsymbol{x}_s; s, t) \right] - h^*(\boldsymbol{v}_\psi(\boldsymbol{x}_s; s, t)) \right], \tag{26}$$

By the trick on the score function $\nabla \log q_\phi(\boldsymbol{x}_s; s, t)$, we have

$$\mathbb{E}_{q_\phi(\boldsymbol{x}_s; s, t)} \langle \boldsymbol{v}_\psi(\boldsymbol{x}_s; s, t), \nabla \log q_\phi(\boldsymbol{x}_s; s, t) \rangle = \mathbb{E}_{q_\phi(\boldsymbol{x}_s, \boldsymbol{x}_t; s, t)} \langle \boldsymbol{v}_\psi(\boldsymbol{x}_s; s, t), \nabla \log q_\phi(\boldsymbol{x}_s | \boldsymbol{x}_t; s, t) \rangle. \tag{27}$$

Therefore, the optimization problem (25) can be rewritten as these two stages optimization problem

$$\min_{q_\phi} \quad \mathbb{E}_{q_\phi(\boldsymbol{x}_s, \boldsymbol{x}_t; s, t)} \left[ \boldsymbol{v}_\psi(\boldsymbol{x}_s; s, t)^T \left[ \nabla \log p(\boldsymbol{x}_s; s) - \nabla_{\boldsymbol{x}_s} \log q_\phi(\boldsymbol{x}_s | \boldsymbol{x}_t; s, t) \right] - h^*(\boldsymbol{v}_\psi(\boldsymbol{x}_s; s, t)) \right], \tag{28}$$

$$\min_{\boldsymbol{v}_\psi} \quad \mathbb{E}_{q_\phi(\boldsymbol{x}_s, \boldsymbol{x}_t; s, t)} \left[ \boldsymbol{v}_\psi(\boldsymbol{x}_s; s, t)^T \left[ \nabla_{\boldsymbol{x}_s} \log q_\phi(\boldsymbol{x}_s | \boldsymbol{x}_t; s, t) - \nabla \log p(\boldsymbol{x}_s; s) \right] + h^*(\boldsymbol{v}_\psi(\boldsymbol{x}_s; s, t)) \right]. \tag{29}$$

If the regularization strength parameter $\lambda > 0$, the optimization problem with a regularized term is formed as

$$\min_{\boldsymbol{v}_\psi} \quad \mathbb{E}_{q_\phi(\boldsymbol{x}_s; s, t)} \left[ \boldsymbol{v}_\psi(\boldsymbol{x}_s; s, t)^T \left[ \nabla_{\boldsymbol{x}_s} \log q_\phi(\boldsymbol{x}_s; s, t) - \nabla \log p(\boldsymbol{x}_s; s) \right] + (1 + \lambda) h^*(\boldsymbol{v}_\psi(\boldsymbol{x}_s; s, t)) \right]. \tag{30}$$

We can view the above minization formulation as the Legendre transformation with the convex function $(1 + \lambda) h^*(\cdot)$, then the optimal function $\boldsymbol{v}_{\psi^*(\phi)}(\cdot; s, t)$ of the above optimization problem satisfies

$$\mathbb{E}_{q_\phi(\boldsymbol{x}_s; s, t)} \left[ \boldsymbol{v}_{\psi^*(\phi)}(\cdot; s, t)^T \left[ \nabla_{\boldsymbol{x}_s} \log q_\phi(\boldsymbol{x}_s; s, t) - \nabla \log p(\boldsymbol{x}_s; s) \right] + (1 + \lambda) h^*(\boldsymbol{v}_{\psi^*(\phi)}(\cdot; s, t)) \right], \tag{31}$$

$$= - \mathbb{E}_{q_\phi(\boldsymbol{x}_s; s, t)} \left[ (1 + \lambda) h^* \right]^* \left( \nabla \log p(\boldsymbol{x}_s; s) - \nabla_{\boldsymbol{x}_s} \log q_\phi(\boldsymbol{x}_s; s, t) \right), \tag{32}$$

$$= - \mathbb{E}_{q_\phi(\boldsymbol{x}_s; s, t)} (1 + \lambda) h \left( \frac{1}{1 + \lambda} (\log p(\boldsymbol{x}_s; s) - \nabla_{\boldsymbol{x}_s} \log q_\phi(\boldsymbol{x}_s; s, t)) \right). \tag{33}$$

Bring the above optimal function $\boldsymbol{v}_{\psi^*(\phi)}(\cdot; s, t)$ back to the first stage optimization problem (28), then we have

$$\mathbb{E}_{q_\phi(\boldsymbol{x}_s, \boldsymbol{x}_t; s, t)} \left[ \boldsymbol{v}_{\psi^*(\phi)}(\boldsymbol{x}_s; s, t)^T \left[ \nabla \log p(\boldsymbol{x}_s; s) - \nabla_{\boldsymbol{x}_s} \log q_\phi(\boldsymbol{x}_s | \boldsymbol{x}_t; s, t) \right] - h^*(\boldsymbol{v}_{\psi^*(\phi)}(\boldsymbol{x}_s; s, t)) \right], \tag{34}$$

$$= \mathbb{E}_{q_\phi(\boldsymbol{x}_s, \boldsymbol{x}_t; s, t)} (1 + \lambda) h \left( \frac{1}{1 + \lambda} (\log p(\boldsymbol{x}_s; s) - \nabla_{\boldsymbol{x}_s} \log q_\phi(\boldsymbol{x}_s; s, t)) \right) + \lambda h^* \left( \boldsymbol{v}_{\psi^*(\phi)}(\boldsymbol{x}_s; s, t) \right). \tag{35}$$

As $h$ is a strongly smooth convex function, the Legendre transformation $h^*$ is also a strongly convex function, and $h(\boldsymbol{x}) = 0$ if and only if $\boldsymbol{x} = \boldsymbol{0}$. When $\boldsymbol{v}_{\psi^*(\phi)}(\boldsymbol{x}_s; s, t) \equiv \boldsymbol{0}$, the first derivate condition of the optimal function $\boldsymbol{v}_{\psi^*(\phi)}(\cdot; s, t)$ is satisfied

$$[\nabla_{\boldsymbol{x}_s} \log q_\phi(\boldsymbol{x}_s; s, t) - \nabla \log p(\boldsymbol{x}_s; s)] + (1 + \lambda)h(\boldsymbol{0}) = [\nabla_{\boldsymbol{x}_s} \log q_\phi(\boldsymbol{x}_s; s, t) - \nabla \log p(\boldsymbol{x}_s; s)] = 0.$$

Therefore, the global optimal distribution of (34) is $q_{\phi^*}(\boldsymbol{x}_s; s, t) = \nabla \log p(\boldsymbol{x}_s; s)$ that similar with the original optimization problem (25). So the two stage optimization problem with regularization term has the similar optimal solution as the original optimization problem.

In practice, we choose the function $h(\boldsymbol{x}) = \frac{1}{4\alpha}\|\boldsymbol{x}\|_2^2$, then the Legendre transformation of $h(\boldsymbol{x})$ is that

$$h^*(\boldsymbol{y}) = \max_{\boldsymbol{x}}\{\boldsymbol{y}^T\boldsymbol{x} - \frac{1}{4\alpha}\|\boldsymbol{x}\|_2^2\} = \alpha\|\boldsymbol{y}\|_2^2.$$

Substituting the Legendre transformation $h^*(\boldsymbol{x})$ back into the second stage optimization problem (30), we obtain

$$\mathbb{E}_{q_\phi(\boldsymbol{x}_s; s, t)}\left[\boldsymbol{v}_\psi(\boldsymbol{x}_s; s, t)^T\left[\nabla \log q_\phi(\boldsymbol{x}_s; s, t) - \nabla \log p(\boldsymbol{x}_s; s)\right] + (1 + \lambda)\alpha\|(\boldsymbol{v}_\psi(\boldsymbol{x}_s; s, t)\|_2^2\right],$$

$$= (1 + \lambda)\alpha\mathbb{E}_{q_\phi(\boldsymbol{x}_s; s, t)}\left\|\boldsymbol{v}_\psi(\boldsymbol{x}_s; s, t) - \frac{1}{2(1 + \lambda)\alpha}\left[-\nabla \log q_\phi(\boldsymbol{x}_s; s, t) + \nabla \log p(\boldsymbol{x}_s; s)\right]\right\|_2^2 + C(\phi),$$

where $C(\phi) := \mathbb{E}_{q_\phi(\boldsymbol{x}_s; s, t)}\|\nabla \log q_\phi(\boldsymbol{x}_s; s, t) - \nabla \log p(\boldsymbol{x}_s; s)\|_2^2$ and is independent of $\psi$. Therefore, the optimal function of the second stage optimization problem (30) is that

$$\boldsymbol{v}_{\psi^*(\phi)}(\boldsymbol{x}_s; s, t) = \nabla \log p(\boldsymbol{x}_s; s) - \boldsymbol{f}_{\psi^*(\phi)}(\boldsymbol{x}_s; s, t)$$
$$= \frac{1}{2(1 + \lambda)\alpha}\left[-\nabla \log q_\phi(\boldsymbol{x}_s; s, t) + \nabla \log p(\boldsymbol{x}_s; s)\right].$$

Replacing the target score function $\nabla \log p(\boldsymbol{x}_s; s)$ with the estimated score model $\boldsymbol{S}_{\theta^*}(\boldsymbol{x}_s; s)$, we eventually write the optimal $\boldsymbol{f}_{\tilde{\psi}^*(\phi)}(\boldsymbol{x}_s; s, t)$ of the second stage optimization problem with the regularization term as

$$\boldsymbol{u}_{\tilde{\psi}^*(\phi)}(\boldsymbol{x}_s; s, t) = \boldsymbol{S}_{\theta^*}(\boldsymbol{x}_s; s) - \boldsymbol{f}_{\tilde{\psi}^*(\phi)}(\boldsymbol{x}_s; s, t) \tag{36}$$

$$\boldsymbol{f}_{\tilde{\psi}^*(\phi)}(\boldsymbol{x}_s; s, t) = \frac{1}{2\alpha(1 + \lambda)}\nabla \log q_\phi(\boldsymbol{x}_s; s, t) + (1 - \frac{1}{2\alpha(1 + \lambda)})\boldsymbol{S}_{\theta^*}(\boldsymbol{x}_s; s). \tag{37}$$

Moreover, (34) is rewritten as

$$\mathbb{E}_{q_\phi(\boldsymbol{x}_s, \boldsymbol{x}_t; s, t)}\left[\boldsymbol{u}_{\tilde{\psi}^*(\phi)}(\boldsymbol{x}_s; s, t)^T\left[\boldsymbol{S}_{\theta^*}(\boldsymbol{x}_s; s) - \nabla_{\boldsymbol{x}_s} \log q_\phi(\boldsymbol{x}_s|\boldsymbol{x}_t; s, t)\right] - \alpha\|(\boldsymbol{u}_{\tilde{\psi}^*(\phi)}(\boldsymbol{x}_s; s, t))\|_2^2\right],$$

$$= \frac{(1 + 2\lambda)}{4\alpha(1 + \lambda)^2}\mathbb{E}_{q_\phi(\boldsymbol{x}_s; s, t)}\|\boldsymbol{S}_{\theta^*}(\boldsymbol{x}_s; s) - \nabla \log q_\phi(\boldsymbol{x}_s; s, t)\|_2^2.$$

Therefore, the equilibrium of the two-stage optimization problem for $\phi$ remains consistent with (11).

$\square$

### B.2. Proof of Proposition 3.2

*Proof.* As discussed in the proof of theorem 3.1, the two-stages optimization problem in (15) is equivalent to optimizing

$$\min_\phi \frac{(1 + 2\lambda)}{4\alpha(1 + \lambda)^2}\mathbb{E}_{q_\phi(\boldsymbol{x}_s; s, t)}\|\nabla \log p(\boldsymbol{x}_s; s) - \nabla \log q_\phi(\boldsymbol{x}_s; s, t)\|_2^2. \tag{38}$$

And the continuous transition kernel $q_{\text{trans}}(\boldsymbol{x}_s|\boldsymbol{x}_t; s, t)$ is parameterized as

$$q_\phi(\boldsymbol{x}_s|\boldsymbol{x}_t; s, t) = \mathcal{N}(\boldsymbol{x}_s; a(s)\boldsymbol{G}_\phi(\boldsymbol{x}_t, t), \sigma(s)^2\boldsymbol{I}), \tag{39}$$

where $0 < s < t \leq T$, $a(s), \sigma(s) \in \mathbb{R}^+$ are defined in (2). Then we have that the marginal distribution $q_\phi(\boldsymbol{x}_s; s, t)$ is supported on $\mathbb{R}^d$ since

$$q_\phi(\boldsymbol{x}_s; s, t) = \int q_\phi(\boldsymbol{x}_s|\boldsymbol{x}_t; s, t)p(\boldsymbol{x}_t; t)d\boldsymbol{x}_t > 0, \quad \forall \boldsymbol{x}_s \in \mathbb{R}^d. \tag{40}$$

Therefore, the optimal variational distribution $q_\phi(\boldsymbol{x}_s; s, t)$ in (38) satisfies that

$$\nabla \log p(\boldsymbol{x}_s; s) = \nabla \log q_\phi(\boldsymbol{x}_s; s, t), \quad \forall \boldsymbol{x}_s \in \mathbb{R}^d.$$

It implies that $q_\phi(\boldsymbol{x}_s; s, t) = p(\boldsymbol{x}_s; s)$. Furthermore, for the independent random variables $\boldsymbol{x}_t \sim p(\cdot; t)$ and $\boldsymbol{\epsilon} \sim \mathcal{N}(\boldsymbol{0}, \boldsymbol{I})$, the characteristic function of $\boldsymbol{G}_\phi(\cdot, t) \sharp p(\cdot; t)$ satisfies

$$\begin{aligned}
\varphi_{\boldsymbol{G}_\phi(\cdot,t) \sharp p(\cdot;t)}(\boldsymbol{w}) &= \mathbb{E} e^{i \boldsymbol{w}^T \boldsymbol{G}_\phi(\boldsymbol{x}_t, t)} = \mathbb{E} e^{i \frac{1}{a(s)} \boldsymbol{w}^T [a(s) \boldsymbol{G}_\phi(\boldsymbol{x}_t, t) + \sigma(s) \boldsymbol{\epsilon}]} (\mathbb{E} e^{i \frac{\sigma(s)}{a(s)} \boldsymbol{w}^T \boldsymbol{\epsilon}})^{-1}, \\
&= \mathbb{E}_{q_\phi(\boldsymbol{x}_s; s, t)} e^{i \frac{1}{a(s)} \boldsymbol{w}^T \boldsymbol{x}_s} (\mathbb{E} e^{i \frac{\sigma(s)}{a(s)} \boldsymbol{w}^T \boldsymbol{\epsilon}})^{-1} = \mathbb{E}_{p(\boldsymbol{x}_s; s)} e^{i \frac{1}{a(s)} \boldsymbol{w}^T \boldsymbol{x}_s} (\mathbb{E} e^{i \frac{\sigma(s)}{a(s)} \boldsymbol{w}^T \boldsymbol{\epsilon}})^{-1}, \\
&= \mathbb{E}_{p(\boldsymbol{x}_0; 0) \cdot \mathcal{N}(\boldsymbol{\epsilon}; \boldsymbol{0}, \boldsymbol{I})} e^{i \frac{1}{a(s)} \boldsymbol{w}^T [a(s) \boldsymbol{x}_0 + \sigma(s) \boldsymbol{\epsilon}]} (\mathbb{E} e^{i \frac{\sigma(s)}{a(s)} \boldsymbol{w}^T \boldsymbol{\epsilon}})^{-1} = \varphi_{\boldsymbol{x}_0}(\boldsymbol{w}).
\end{aligned}$$

Therefore, the characteristic function of $\boldsymbol{G}_\phi(\cdot, t) \sharp p(\cdot; t)$ is the same as the characteristic function of $p(\boldsymbol{x}_0; 0)$. We can conclude that $\boldsymbol{G}_\phi(\cdot, t)$ establishes a mapping from distribution $p(\cdot; t)$ back to $p(\cdot; 0)$. $\qquad \square$

### B.3. Proof of Proposition 3.4

*Proof of Proposition 3.4.* As the approximated variational distribution $q_{\hat{\phi}}$ is obtained by

$$q_{\hat{\phi}}(\cdot; s, t) \in \underset{q_\phi \in \mathcal{Q}}{\arg \min} \, \mathbb{E}_{q_\phi(\boldsymbol{x}_s, \boldsymbol{x}_t; s, t)} \Big[ \boldsymbol{u}_{\hat{\psi}(\phi)}(\boldsymbol{x}_s; s, t)^T \left[ \nabla \log p(\boldsymbol{x}_s; s) - \nabla_{\boldsymbol{x}_s} \log q_\phi(\boldsymbol{x}_s | \boldsymbol{x}_t; s, t) \right] - \alpha \| \boldsymbol{u}_{\hat{\psi}(\phi)}(\boldsymbol{x}_s; s, t) \|_2^2 \Big], \tag{41}$$

where $\boldsymbol{u}_{\hat{\psi}(\phi)}(\boldsymbol{x}_s; s, t) := \nabla \log p(\boldsymbol{x}_s; s) - \boldsymbol{f}_{\hat{\psi}(\phi)}(\boldsymbol{x}_s; s, t)$.

We can bound the Fisher divergence between $q_{\hat{\phi}}$ and $p(\cdot; s)$ as follows. First we rewrite the first stage optimization objective as

$$\begin{aligned}
&\mathbb{E}_{q_\phi(\boldsymbol{x}_s, \boldsymbol{x}_t; s, t)} \Big[ \boldsymbol{u}_\psi(\boldsymbol{x}_s; s, t)^T \left[ \nabla \log p(\boldsymbol{x}_s; s) - \nabla_{\boldsymbol{x}_s} \log q_\phi(\boldsymbol{x}_s | \boldsymbol{x}_t; s, t) \right] - \alpha \| \boldsymbol{u}_\psi(\boldsymbol{x}_s; s, t) \|_2^2 \Big], \\
=&\mathbb{E}_{q_\phi(\boldsymbol{x}_s; s, t)} \Big[ \boldsymbol{u}_\psi(\boldsymbol{x}_s; s, t)^T \left[ \nabla \log p(\boldsymbol{x}_s; s) - \nabla_{\boldsymbol{x}_s} \log q_\phi(\boldsymbol{x}_s; s, t) \right] - \alpha \| \boldsymbol{u}_\psi(\boldsymbol{x}_s; s, t) \|_2^2 \Big], \\
=&\frac{1 + 2\lambda}{4\alpha(1 + \lambda)^2} \mathbb{E}_{q_\phi(\boldsymbol{x}_s; s, t)} \| \boldsymbol{\delta}_\phi(\boldsymbol{x}_s; s, t) \|_2^2 + \frac{\lambda}{1 + \lambda} \mathbb{E}_{q_\phi(\boldsymbol{x}_s; s, t)} \left\langle \boldsymbol{\delta}_\phi(\boldsymbol{x}_s; s, t), \boldsymbol{u}_\psi(\boldsymbol{x}_s; s, t) - \frac{1}{2\alpha(1 + \lambda)} \boldsymbol{\delta}_\phi(\boldsymbol{x}_s; s, t) \right\rangle \\
&-\alpha \mathbb{E}_{q_\phi(\boldsymbol{x}_s; s, t)} \| \boldsymbol{u}_\psi(\boldsymbol{x}_s; s, t) - \frac{1}{2\alpha(1 + \lambda)} \boldsymbol{\delta}_\phi(\boldsymbol{x}_s; s, t) \|_2^2, \tag{42}
\end{aligned}$$

where $\boldsymbol{\delta}_\phi(\boldsymbol{x}_s; s, t) := \nabla \log p(\boldsymbol{x}_s; s) - \nabla \log q_\phi(\boldsymbol{x}_s; s, t)$. Then we have

$$\begin{aligned}
\frac{1}{4\alpha(1 + \lambda)^2} \mathrm{FI}(q_{\hat{\phi}}(\cdot; s, t) \| p(\cdot; s)) &= \frac{1}{4\alpha(1 + \lambda)^2} \mathbb{E}_{q_{\hat{\phi}}(\boldsymbol{x}_s; s, t)} \| \nabla \log p(\boldsymbol{x}_s; s) - \nabla \log q_{\hat{\phi}}(\boldsymbol{x}_s; s, t) \|_2^2 \\
&= \underbrace{\frac{1 + 2\lambda}{4\alpha(1 + \lambda)^2} \mathbb{E}_{q_{\hat{\phi}}(\boldsymbol{x}_s; s, t)} \| \boldsymbol{\delta}_{\hat{\phi}}(\boldsymbol{x}_s; s, t) \|_2^2}_{①} - \frac{\lambda}{2\alpha(1 + \lambda)^2} \mathrm{FI}(q_{\hat{\phi}}(\cdot; s, t) \| p(\cdot; s)),
\end{aligned}$$

Bring (42) into ①, we have

$$\begin{aligned}
① =&\mathbb{E}_{q_{\hat{\phi}}(\boldsymbol{x}_s; s, t)} \Big\{ \Big[ \boldsymbol{u}_{\hat{\psi}(\hat{\phi})}(\boldsymbol{x}_s; s, t)^T \left[ \nabla \log p(\boldsymbol{x}_s; s) - \nabla_{\boldsymbol{x}_s} \log q_{\hat{\phi}}(\boldsymbol{x}_s; s, t) \right] - \alpha \| \boldsymbol{u}_{\hat{\psi}(\hat{\phi})}(\boldsymbol{x}_s; s, t) \|_2^2 \Big] \\
&\underbrace{- \frac{\lambda}{1 + \lambda} \left\langle \boldsymbol{\delta}_{\hat{\phi}}(\boldsymbol{x}_s; s, t), \boldsymbol{u}_{\hat{\psi}(\hat{\phi})}(\boldsymbol{x}_s; s, t) - \frac{1}{2\alpha(1 + \lambda)} \boldsymbol{\delta}_{\hat{\phi}}(\boldsymbol{x}_s; s, t) \right\rangle + \alpha \| \boldsymbol{u}_{\hat{\psi}(\hat{\phi})}(\boldsymbol{x}_s; s, t) - \frac{1}{2\alpha(1 + \lambda)} \boldsymbol{\delta}_{\hat{\phi}}(\boldsymbol{x}_s; s, t) \|_2^2}_{②} \Big\}, \\
\leq&\underbrace{\mathbb{E}_{q_{\phi^*}(\boldsymbol{x}_s; s, t)} \Big\{ \Big[ \boldsymbol{u}_{\hat{\psi}(\phi^*)}(\boldsymbol{x}_s; s, t)^T \left[ \nabla \log p(\boldsymbol{x}_s; s) - \nabla \log q_{\phi^*}(\boldsymbol{x}_s; s, t) \right] - \alpha \| \boldsymbol{u}_{\hat{\psi}(\phi^*)}(\boldsymbol{x}_s; s, t) \|_2^2 \Big]}_{③} - \mathbb{E}_{q_{\hat{\phi}}(\boldsymbol{x}_s; s, t)}(②) + \alpha \varepsilon_f^2.
\end{aligned}$$

For the term $\mathbb{E}_{q_{\hat{\phi}}(\boldsymbol{x}_s;s,t)}(②)$, we have

$$-\mathbb{E}_{q_{\hat{\phi}}(\boldsymbol{x}_s;s,t)}(②) \leq \frac{\lambda}{2\alpha(1+\lambda)^2}\mathbb{E}_{q_{\hat{\phi}}(\boldsymbol{x}_s;s,t)}\|\boldsymbol{\delta}_{\hat{\phi}}(\boldsymbol{x}_s;s,t)\|_2^2 + \frac{\alpha\lambda}{2}\mathbb{E}_{q_{\hat{\phi}}(\boldsymbol{x}_s;s,t)}\|\boldsymbol{u}_{\hat{\psi}(\hat{\phi})}(\boldsymbol{x}_s;s,t) - \frac{1}{2\alpha(1+\lambda)}\boldsymbol{\delta}_{\hat{\phi}}(\boldsymbol{x}_s;s,t)\|_2^2,$$

$$\leq \frac{\lambda}{2\alpha(1+\lambda)^2}\mathrm{FI}(q_{\hat{\phi}}(\cdot;s,t)\|p(\cdot;s)) + \frac{\alpha\lambda}{2}\varepsilon_{\mathrm{f}}^2.$$

For the term ③, by (42)we have

$$③ = \frac{1+2\lambda}{4\alpha(1+\lambda)^2}\mathbb{E}_{q_{\phi^*}(\boldsymbol{x}_s;s,t)}\|\boldsymbol{\delta}_{\phi^*}(\boldsymbol{x}_s;s,t)\|_2^2 - \alpha\mathbb{E}_{q_{\phi^*}(\boldsymbol{x}_s;s,t)}\|\boldsymbol{u}_{\hat{\psi}(\phi^*)}(\boldsymbol{x}_s;s,t) - \frac{1}{2\alpha(1+\lambda)}\boldsymbol{\delta}_{\phi^*}(\boldsymbol{x}_s;s,t)\|_2^2$$

$$+ \frac{\lambda}{1+\lambda}\mathbb{E}_{q_{\phi^*}(\boldsymbol{x}_s;s,t)}\left\langle \boldsymbol{\delta}_{\phi^*}(\boldsymbol{x}_s;s,t), \boldsymbol{u}_{\hat{\psi}(\phi^*)}(\boldsymbol{x}_s;s,t) - \frac{1}{2\alpha(1+\lambda)}\boldsymbol{\delta}_{\phi^*}(\boldsymbol{x}_s;s,t)\right\rangle,$$

$$\leq \frac{1+2\lambda}{4\alpha(1+\lambda)^2}\mathrm{FI}(q_{\phi^*}(\cdot;s,t)\|p(\cdot;s)) - \alpha\mathbb{E}_{q_{\phi^*}(\boldsymbol{x}_s;s,t)}\|\boldsymbol{u}_{\hat{\psi}(\phi^*)}(\boldsymbol{x}_s;s,t) - \frac{1}{2\alpha(1+\lambda)}\boldsymbol{\delta}_{\phi^*}(\boldsymbol{x}_s;s,t)\|_2^2$$

$$+ \frac{\lambda^2}{4(1+\lambda)^2\alpha}\mathbb{E}_{q_{\phi^*}(\boldsymbol{x}_s;s,t)}\|\boldsymbol{\delta}_{\phi^*}(\boldsymbol{x}_s;s,t)\|_2^2 + \alpha\mathbb{E}_{q_{\phi^*}(\boldsymbol{x}_s;s,t)}\|\boldsymbol{u}_{\hat{\psi}(\phi^*)}(\boldsymbol{x}_s;s,t) - \frac{1}{2\alpha(1+\lambda)}\boldsymbol{\delta}_{\phi^*}(\boldsymbol{x}_s;s,t)\|_2^2,$$

$$= \frac{1}{4\alpha}\mathrm{FI}(q_{\phi^*}(\cdot;s,t)\|p(\cdot;s)).$$

Bring the above results back to ①, we have

$$\mathrm{FI}(q_{\hat{\phi}}(\cdot;s,t)\|p(\cdot;s)) = 4\alpha(1+\lambda)^2(① - \frac{\lambda}{2\alpha(1+\lambda)^2}\mathrm{FI}(q_{\hat{\phi}}(\cdot;s,t)\|p(\cdot;s))),$$

$$\leq (1+\lambda)^2\mathrm{FI}(q_{\phi^*}(\cdot;s,t)\|p(\cdot;s)) + 2\alpha^2(1+\lambda)^2(2+\lambda)\varepsilon_{\mathrm{f}}^2. \tag{43}$$

Introduce the approximation gap from Assumption 3.5, we have

$$\mathrm{FI}(q_{\hat{\phi}}(\cdot;s,t)\|p(\cdot;s)) \leq 2(1+\lambda)^2\varepsilon_{\mathrm{c}}^2\mathrm{FI}(p(\cdot;t+s)\|p(\cdot,s)) + 2\alpha^2(1+\lambda)^2(2+\lambda)\varepsilon_{\mathrm{f}}^2. \tag{44}$$

The assumption 3.5 quantifies the capacity of the consistency map $\boldsymbol{G}_\phi(\cdot,t)$ based on the initial gap. That is, the $q_{\phi_0}(\boldsymbol{x};s,t) = \int p(\boldsymbol{x}|\boldsymbol{x}_t;0,s)p(\boldsymbol{x}_t;t)\mathrm{d}\boldsymbol{x}_t = p(\boldsymbol{x};t+s)$ when $\boldsymbol{G}_{\phi_0}(\boldsymbol{x}_t,t) \equiv \boldsymbol{x}_t$. Therefore, the initial approximation gap $\mathrm{FI}(q_{\phi_0}(\cdot;t+s)\|p(\cdot,s)) = \mathrm{FI}(p(\cdot;t+s)\|p(\cdot,s))$. $\qquad\square$

## B.4. Proof of Proposition 3.8

*Proof of Proposition 3.8.* First we bound the Wasserstein distance between $q_{\hat{\phi}}(\cdot;0,t)$ and $p(\cdot;0)$. Since $p(\cdot;0)$ is supported on the hypercube $[-1,1]^d$, its support is contained in the Euclidean ball $B(0,\sqrt{d})$. By the results of bounds on the logarithmic Sobolev inequality (LSI) constants (Bardet et al., 2018; Chen et al., 2021; Cattiaux & Guillin, 2022), we can derive the following LSI inequality

$$\mathbb{E}_{p(\boldsymbol{x}_\delta;\delta)}f^2(\boldsymbol{x}_\delta)(\log f^2(\boldsymbol{x}_\delta) - \log\mathbb{E}_{p(\boldsymbol{x}_\delta;\delta)}f^2(\boldsymbol{x}_\delta)) \leq 2C_{\mathrm{LSI}}\mathbb{E}_{p(\boldsymbol{x}_\delta;\delta)}\|\nabla f(\boldsymbol{x}_\delta)\|_2^2, \quad \forall f \in \mathcal{C}^\infty(\mathbb{R}^d), \tag{45}$$

where the LSI constant $C_{\mathrm{LSI}} \leq 6(4d + \sigma(\delta))\exp(\frac{4d}{\sigma(\delta)^2})$. Let $f^2(\cdot) = \frac{q_{\hat{\phi}}(\cdot;\delta,t)}{p(\cdot;\delta)}$, then we have

$$\mathrm{KL}(q_{\hat{\phi}}(\cdot;\delta,t)\|p(\cdot;\delta)) \lesssim (4d + \sigma(\delta))\exp(\frac{4d}{\sigma(\delta)^2})\mathrm{FI}(q_{\hat{\phi}}(\cdot;\delta,t)\|p(\cdot;\delta))$$

Moreover, by Otto-Villani theorem (Otto & Villani, 2000), the following Talagrand's inequality holds

$$W_2^2(q_{\hat{\phi}}(\cdot;\delta,t), p(\cdot;\delta)) \leq 2C_{\mathrm{LSI}}\mathrm{KL}(q_{\hat{\phi}}(\cdot;\delta,t)\|p(\cdot;\delta)) \lesssim (4d + \sigma(\delta))^2\exp(\frac{8d}{\sigma(\delta)^2})\mathrm{FI}(q_{\hat{\phi}}(\cdot;\delta,t)\|p(\cdot;\delta)).$$

Denote $\Delta := (4d + \sigma(\delta))^2\exp(\frac{8d}{\sigma(\delta)^2})$, we have the bound of Wasserstein distance between $q_{\hat{\phi}}(\cdot;\delta,t)$ and $p(\cdot;\delta)$ as

$$W_2^2(q_{\hat{\phi}}(\cdot;\delta,t), p(\cdot;\delta)) \lesssim \mathcal{E}_{W_2^2}(t) := \varepsilon_{\mathrm{c}}^2\Delta\mathrm{FI}(p(\cdot;t+\delta)\|p(\cdot,\delta)) + \Delta\varepsilon_{\mathrm{f}}^2. \tag{46}$$

Next, following the methodology proposed in (Lyu et al., 2024), we give the bound for the multistep sampling of CoSIM. Given a sequence of fixed time points $T = t_0 \geq t_1 \geq \cdots \geq t_{K-1} \geq t_K = \delta$, we denote $\boldsymbol{z}_0 \sim p(\cdot; t_0)$ and

$$\boldsymbol{z}_k = a(t_k)\boldsymbol{G}_{\hat{\phi}}(\boldsymbol{z}_{k-1}, t_{k-1}) + \sigma(t_k)\boldsymbol{\epsilon}_{2k}, \tag{47}$$

$$q_{\hat{\phi}}^{(k)}(\cdot; \delta) = \text{Law}\{\boldsymbol{z}_k^{(\delta)} := a(\delta)\boldsymbol{G}_{\hat{\phi}}(\boldsymbol{z}_{k-1}, t_{k-1}) + \sigma(\delta)\boldsymbol{\epsilon}_{2k-1}\}, \quad k = 1, 2, \cdots, K, \tag{48}$$

where $\boldsymbol{\epsilon}_k \sim \mathcal{N}(\boldsymbol{0}, \boldsymbol{I})$ are independent Gaussian noises, then it can be shown that $q_{\hat{\phi}}^{(1)}(\cdot; \delta) = q_{\hat{\phi}}(\cdot; \delta, T)$. For a fixed time points $t_k$, we consider the optimal transport coupling $\gamma(\boldsymbol{z}_k, \boldsymbol{x}_{t_k}) \in \Gamma_{\text{opt}}(\text{Law}(\boldsymbol{z}_k), p(\cdot; t_k))$ and let

$$\boldsymbol{z}_{k+1}^\delta = \left[a(\delta)\boldsymbol{G}_{\hat{\phi}}(\boldsymbol{z}_k, t_k) + \sigma(\delta)\boldsymbol{\epsilon}_{2k+1}\right] \sim q_{\hat{\phi}}^{(k)}(\cdot; \delta),$$

$$\boldsymbol{y} = \left[a(\delta)\boldsymbol{G}_{\hat{\phi}}(\boldsymbol{x}_{t_k}, t_k) + \sigma(\delta)\boldsymbol{\epsilon}_{2k+1}\right] \sim q_{\hat{\phi}}(\cdot; \delta, t_k).$$

Then we have

$$W_2(q_{\hat{\phi}}^{(k+1)}, p(\cdot; \delta)) \leq W_2(q_{\hat{\phi}}^{(k+1)}, q_{\hat{\phi}}(\cdot; \delta, t_k)) + W_2(q_{\hat{\phi}}(\cdot; \delta, t_k), p(\cdot; \delta)),$$

$$\leq \mathbb{E}_{(\boldsymbol{z}_k, \boldsymbol{x}_{t_k}) \sim \gamma, \boldsymbol{\epsilon}_{2k+1} \sim \mathcal{N}(\boldsymbol{0}, \boldsymbol{I})} \|\boldsymbol{z}_{k+1}^\delta - \boldsymbol{y}\|_2^2 + W_2^2(q_{\hat{\phi}}(\cdot; \delta, t_k), p(\cdot; \delta)),$$

$$= a(\delta)\mathbb{E}_{(\boldsymbol{z}_k, \boldsymbol{x}_{t_k}) \sim \gamma} \|\boldsymbol{G}_{\hat{\phi}}(\boldsymbol{z}_k, t_k) - \boldsymbol{G}_{\hat{\phi}}(\boldsymbol{x}_{t_k}, t_k)\|_2^2 + W_2^2(q_{\hat{\phi}}(\cdot; \delta, t_k), p(\cdot; \delta)). \tag{49}$$

Recall that the consistency function $\boldsymbol{G}_{\hat{\phi}}(\cdot, t)$ is parameterized as

$$\boldsymbol{G}_\phi(\boldsymbol{x}_t, t) = \mathcal{O}\left(c_{\text{skip}}(t)\boldsymbol{x}_t + c_{\text{out}}(t)\boldsymbol{F}_\phi(c_{\text{in}}(t)\boldsymbol{x}_t, t)\right), \tag{50}$$

where

$$c_{\text{skip}}(t) = \frac{\sigma_{\text{data}}^2 a(t)^2}{\sigma(t)^2 + \sigma_{\text{data}}^2 a(t)^2}, \tag{51}$$

$$c_{\text{out}}(t) = \left(\frac{\sigma_{\text{data}}^2 \sigma(t)^2 a(t)^2}{\sigma(t)^2 + \sigma_{\text{data}}^2 a(t)^2}\right)^{1/2}, \tag{52}$$

$$c_{\text{in}}(t) = \left(\frac{1}{\sigma_{\text{data}}^2 a(t)^2 + \sigma(t)^2}\right)^{1/2}. \tag{53}$$

Practically, $\sigma_{\text{data}} := \frac{1}{2}$ is a fixed constant. Given any $\boldsymbol{x}_1, \boldsymbol{x}_2 \in \mathbb{R}^d$ and by the $L_f$-Lipschitz continuity of $\boldsymbol{F}_\phi(\cdot, t)$ with respect to $\boldsymbol{x}_t$, we have

$$\|\boldsymbol{G}_\phi(\boldsymbol{x}_1, t) - \boldsymbol{G}_\phi(\boldsymbol{x}_2, t)\|_2 \leq \|c_{\text{skip}}(t)(\boldsymbol{x}_1 - \boldsymbol{x}_2) + c_{\text{out}}(t)(\boldsymbol{F}_\phi(c_{\text{in}}(t)\boldsymbol{x}_1, t) - \boldsymbol{F}_\phi(c_{\text{in}}(t)\boldsymbol{x}_2, t))\|_2,$$

$$\leq (c_{\text{skip}}(t) + c_{\text{out}}(t)L_f c_{\text{in}}(t))\|\boldsymbol{x}_1 - \boldsymbol{x}_2\|_2,$$

$$\leq \left(\frac{a(t)}{\sigma(t)^2 + a(t)^2} + 2L_f \frac{\sigma(t)a(t)}{\sigma(t)^2 + a(t)^2}\right)\|\boldsymbol{x}_1 - \boldsymbol{x}_2\|_2^2. \tag{54}$$

In the case of variance preserving (VP) scheme for (1), we have

$$a(t) = e^{-t}, \quad \sigma(t) = \sqrt{1 - e^{-2t}}, \quad \forall t \in [0, T]. \tag{55}$$

This implies that $\boldsymbol{G}_\phi(\boldsymbol{x}, t)$ is $3L_f$-Lipschitz continuous with respect to $\boldsymbol{x}$ if $a(t), \sigma(t)$ is defined as (55).

In the case of variance exploding (VE) scheme for (1), we have

$$a(t) = 1, \quad \sigma(t) = t, \quad \forall t \in [0, T]. \tag{56}$$

Then $\boldsymbol{G}_\phi(\boldsymbol{x}, t)$ is $\left(\frac{2L_f}{t+1} + \frac{1}{t^2+1}\right)$-Lipschitz continuous in the variance exploding scenario. Denote these two Lipschitz constant as $L_{\text{vp}} = 3L_f$ adn $L_{\text{ve}} = \frac{2L_f}{t+1} + \frac{1}{t^2+1}$ respectively. To ensure the condition $\sigma(\delta) > a(\delta)\sqrt{d}$, we have the early stopping time $\delta_{\text{vp}} = \mathcal{O}(\log(d))$ for VP scheme and $\delta_{\text{ve}} = \mathcal{O}(\sqrt{d})$ for VE scheme.

For the variance preserving scheme, substituting the above results into (49) yields

$$W_2(q_{\hat{\phi}}^{(k+1)}, p(\cdot; \delta)) \leq a(\delta) L_{\mathrm{vp}} \mathbb{E}_{(\boldsymbol{z}_k, \boldsymbol{x}_{t_k}) \sim \gamma} \|\boldsymbol{z}_k - \boldsymbol{x}_{t_k}\|_2^2 + W_2(q_{\hat{\phi}}(\cdot; \delta, t_k), p(\cdot; \delta)),$$

$$\leq a(\delta) L_{\mathrm{vp}} W_2(\mathrm{Law}(z_k), p(\cdot; t_k)) + W_2(q_{\hat{\phi}}(\cdot; \delta, t_k), p(\cdot; \delta)).$$

Next, given the optimal coupling $\gamma'(\boldsymbol{z}', \boldsymbol{x}') \in \Gamma_{\mathrm{opt}}(q_{\hat{\phi}}^{(k)}(\cdot; \delta), p(\cdot; \delta))$, we consider the following coupling $\gamma_1(\boldsymbol{z}, \boldsymbol{x}) \sim \Gamma(\mathrm{Law}(\boldsymbol{z}_k), p(\cdot; t_k))$

$$\boldsymbol{z} = a(t_k - \delta) \boldsymbol{z}_k^{(\delta)} + \sigma(t_k - \delta) \boldsymbol{\epsilon}_{2k},$$

$$\boldsymbol{x} = a(t_k - \delta) \boldsymbol{x}_\delta + \sigma(t_k - \delta) \boldsymbol{\epsilon}_{2k}.$$

Then

$$W_2(\mathrm{Law}(z_k), p(\cdot; t_k)) \leq (\mathbb{E}_{(\boldsymbol{z}, \boldsymbol{x}) \sim \gamma_1} \|\boldsymbol{z} - \boldsymbol{x}\|_2^2)^{1/2} = a(t_k - \delta) \mathbb{E}_{\gamma'} \|\boldsymbol{z}' - \boldsymbol{x}'\|_2^2 \leq a(t_k - \delta) W_2(q_{\hat{\phi}}^{(k)}(\cdot; \delta), p(\cdot; \delta)).$$

Therefore, we have

$$W_2(q_{\hat{\phi}}^{(k+1)}, p(\cdot; \delta)) \lesssim a(t_k) L_{\mathrm{vp}} W_2(q_{\hat{\phi}}^{(k)}(\cdot; \delta), p(\cdot; \delta)) + \mathcal{E}_{W_2^2}^{\frac{1}{2}}(t_k).$$

Let $t_1 = \cdots = t_{K-1} = t_{\mathrm{mid}} > 1$ with a fixed $t_{\mathrm{mid}}$ and apply the discrete type Grönwall's inequality (Gronwall, 1919), we have

$$W_2(q_{\hat{\phi}}^{(K)}(\cdot; 0), p(\cdot; 0)) \lesssim W_2(q_{\hat{\phi}}^{(K)}(\cdot; \delta), p(\cdot; \delta)) + \delta^2 d$$

$$\lesssim \delta^2 d + (a(t_{\mathrm{mid}}) L_{\mathrm{vp}})^{K-1} \mathcal{E}_{W_2^2}^{\frac{1}{2}}(T) + (1 - a(t_{\mathrm{mid}}) L_{\mathrm{vp}})^{-1} \mathcal{E}_{W_2^2}^{\frac{1}{2}}(t_{\mathrm{mid}}),$$

$$= \delta^2 d + e^{(K-1)(-t_{\mathrm{mid}} + \log(3L_f))} \mathcal{E}_{W_2^2}^{\frac{1}{2}}(T) + (1 - e^{-t_{\mathrm{mid}} + \log(3L_f)})^{-1} \mathcal{E}_{W_2^2}^{\frac{1}{2}}(t_{\mathrm{mid}}).$$

For the variance exploding scheme, we have

$$W_2(q_{\hat{\phi}}^{(K)}(\cdot; 0), p(\cdot; 0)) \lesssim \delta^2 d + (\frac{2L_f}{t_{\mathrm{mid}} + 1})^{K-1} \mathcal{E}_{W_2^2}^{\frac{1}{2}}(T) + (1 - \frac{2L_f}{t_{\mathrm{mid}} + 1})^{-1} \mathcal{E}_{W_2^2}^{\frac{1}{2}}(t_{\mathrm{mid}}).$$

Therefore, the first term on right-hand side has an exponential decay rate as $t_{\mathrm{mid}} = \log(4L_f)$ for VP scheme and $t_{\mathrm{mid}} = 4L_f$ for VE scheme. □

## C. Experimental Details

Our consistency method is to build on top of existing pre-trained diffusion models and distill from them. With numerous existing diffusion models, we choose our base models using the following criteria:

1. Completely open-source, including checkpoints, model architectures, and all training and inferencing details.

2. Results that are generally recognized to be reproducible.

3. State-of-the-art performance.

Fortunately, all these are satisfied with EDM (Karras et al., 2022) and EDM2 (Karras et al., 2024). They are set to be our base models.

**Pre-processing and Evaluation**  We followed EDM to process CIFAR10 and Imagenet $64 \times 64$, and followed EDM2 for Imagenet $512 \times 512$. Most of the consistency methods we compared with in Table 1,2,3 follow the same pre-processing protocol. However, CTM (Kim et al., 2023) followed a different way to down-sample Imagenet to $64 \times 64$. More precisely, CTM has a different down-sampling kernel compared to EDM.

This caussed a significant disruption to the FD-DINOv2 value. Because FD-DINOv2 is computed at a fixed resolution $224 \times 224$ and all images at lower resolution have to be up-sampled before computing this value, the final result will be very sensitive to the down-sampling kernel. For a fair comparison, we compute FD-DINOv2 between CTM generated 50K images and 1,281,167 Imagenet images down-sampled to $64 \times 64$ using the same kernel as CTM.

*Table 5.* Hyperparameters for different experimental setup.

| Hyperparameters | CIFAR10 $32 \times 32$ | Imagenet $64 \times 64$ | Imagenet $512 \times 512$ S | Imagenet $512 \times 512$ M | Imagenet $512 \times 512$ L |
|---|---|---|---|---|---|
| Batch Size | 2048 | 2048 | 2048 | 2048 | 2048 |
| Batch per GPU | 64 | 16 | 32 | 32 | 32 |
| Gradient accumulation round | 4 | 16 | 8 | 8 | 8 |
| # of GPU (L40S 48G) | 8 | 8 | 8 | 8 | 8 |
| Learning rate of $G_\phi$ and $f_\psi$ | $1e^{-5}$ | $4e^{-6}$ | $2e^{-5}$ | $2e^{-5}$ | $1e^{-4}$ |
| # of EMA half-life images | 0.5M | 2M | 2M | 2M | 2M |
| Optimizer Adam eps | $1e^{-8}$ | $1e^{-12}$ | $1e^{-12}$ | $1e^{-12}$ | $1e^{-12}$ |
| Optimizer Adam $\beta_1$ | 0.0 | 0.0 | 0.0 | 0.0 | 0.0 |
| Optimizer Adam $\beta_2$ | 0.999 | 0.99 | 0.99 | 0.99 | 0.99 |
| R | 4 | 8 | 8 | 4 | 4 |
| # of total training images | 200M | 200M | 200M | 200M | 20M |
| # of parameters | 56M | 296M | 280M | 498M | 778M |
| dropout | 0 | 0.1 | 0 | 0.1 | 0.1 |
| augment | 0 | 0 | 0 | 0 | 0 |

*Table 6.* FD-DINOv2 results of CoSIM on class-conditional ImageNet ($512 \times 512$) with different regularization strengths (coef) and total training images. All results are evaluated with NFE= 2.

| Regularization Strength | 204k | 1024k | 2048k | 4096k |
|---|---|---|---|---|
| coef $= 0.5$ | **309.77** | 101.33 | 69.65 | 61.53 |
| coef $= 0.75$ | 392.85 | **92.81** | 61.77 | 50.54 |
| coef $= 1.0$ | 421.90 | 95.23 | **58.63** | **49.28** |

**Network Architecture and Initialization**   Following Algorithm 2 and Section 3.3, there are three networks in our training process, generator $G_\phi$, teacher $S_{\theta^*}$ and auxiliary function $f_\psi$. All are initialized from the pre-trained checkpoints of our chosen base model. During our training, $S_{\theta^*}$ is frozen while generator $G_\phi$ and auxiliary function $f_\psi$ are iteratively refined. During inferencing, only generator $G_\phi$ is used to generate new samples.

Since $G_\phi, f_\psi, S_{\theta^*}$ are initialized from the same checkpoint, their architectures are kept the same as the base model. But $f_\psi$ must accommodate the additional $s$ input from Section 3, so we duplicate the time-embedding layer as described in Section 4. This only adds a very small number of extra parameters during training, and makes no change to the generator used during inferencing.

**Hyperparameters**   The hyperparameters for all of our experiments are presented in Table 5. For those parameters not mentioned in this table, they are kept the same as the base model.

**Training Budget**   We conducted all of our experiments using $8\times$ NVIDIA L40S GPU with 48GB video memory. For CIFAR10 $32 \times 32$, we train our models for $\sim 4$ days. For Imagenet $64 \times 64$, we need $\sim 7$ days to reach our reported results. On Imagenet $512 \times 512$, though the final image size is significantly larger, the training of consistency model is still conducted on $64 \times 64$ internally. This is because EDM2 (Karras et al., 2024) applied a VAE to encode original input size into $64 \times 64$ latent size, and our consistency model only works on the latent feature. For S,M,L setups, the training takes $\sim 3, \sim 7$ and $\sim 3$ days accordingly.

**Sampling Steps**   In Section 4.2, we show that 4-step sampling generates better quality than 2-step sampling. Here we provide more visual results in Figure 8 . Compared to 4-step sampling results, some details deteriorate in 2-step sampling, such as the floating leaves on the spider web, and the shape of shoes are not ideal.

**On the Role of $\lambda$**   We conducted an ablation study on $\lambda$ with the coefficient $\text{coef} := \alpha(1 + \lambda)$ in (15) for ImageNet $512 \times 512$ generation of CoSIM using the $L$ model size with fixed $\alpha = 1.2$.

The results show that introducing regularization (i.e., increasing coef) within a suitable range significantly enhances the learning process of $f_\psi$, which in turn improves the training of $q_\phi$.

Here we provide more visual results on our experiments.

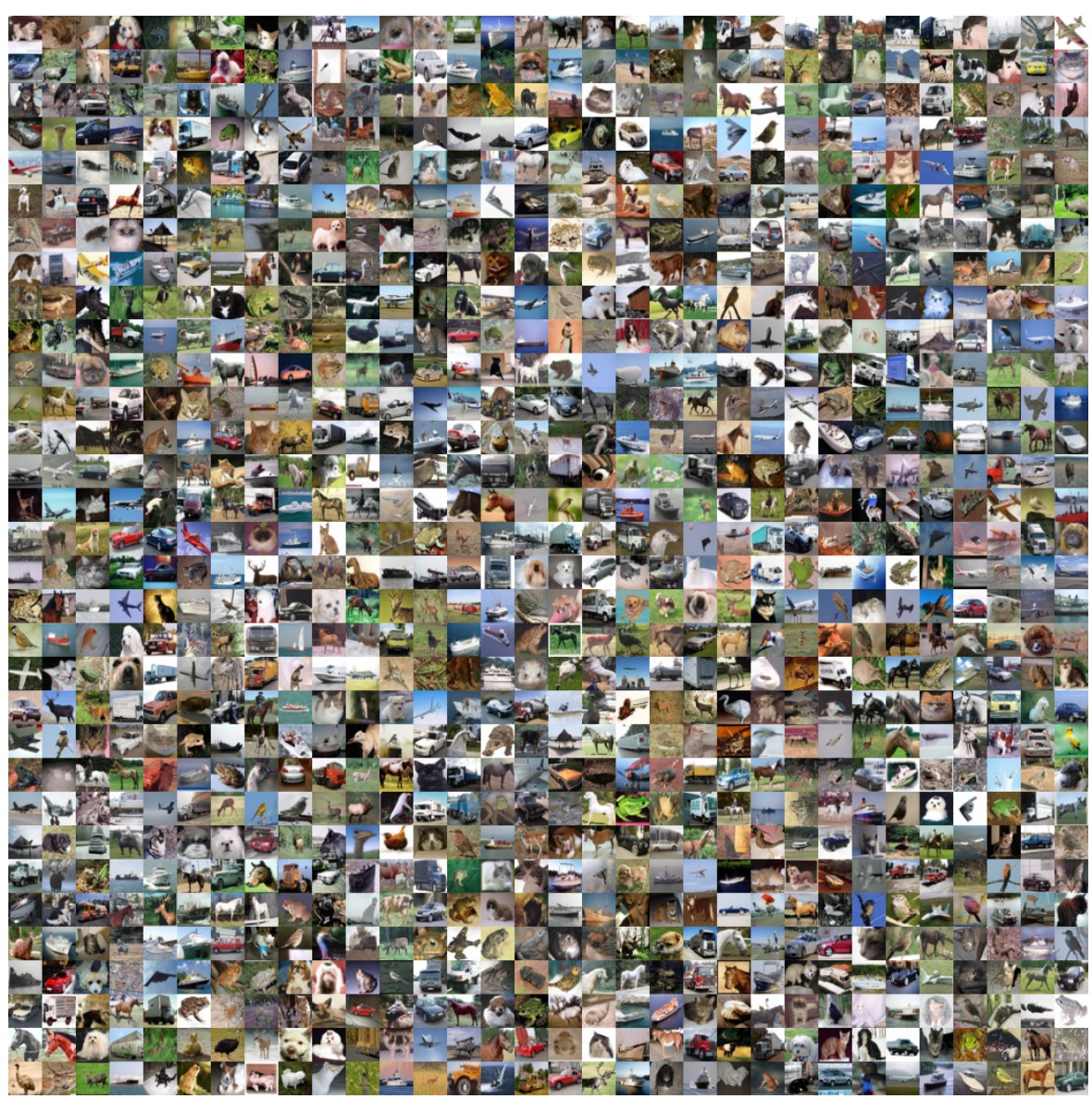

*Figure 3.* Unconditionally generated $32 \times 32$ images on CIFAR10 using 2-step sampling.

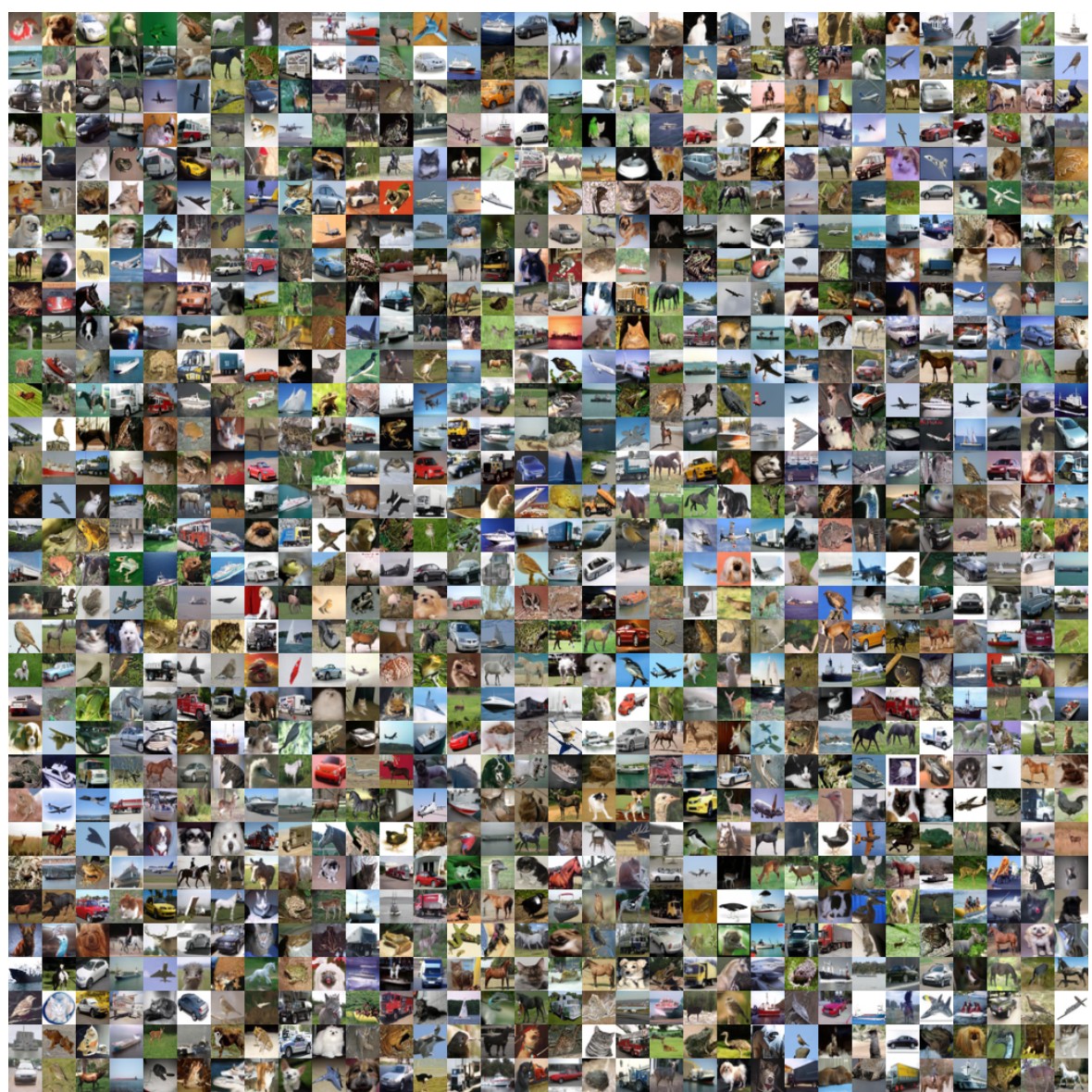

*Figure 4.* Unconditionally generated $32 \times 32$ images on CIFAR10 using 4-step sampling.

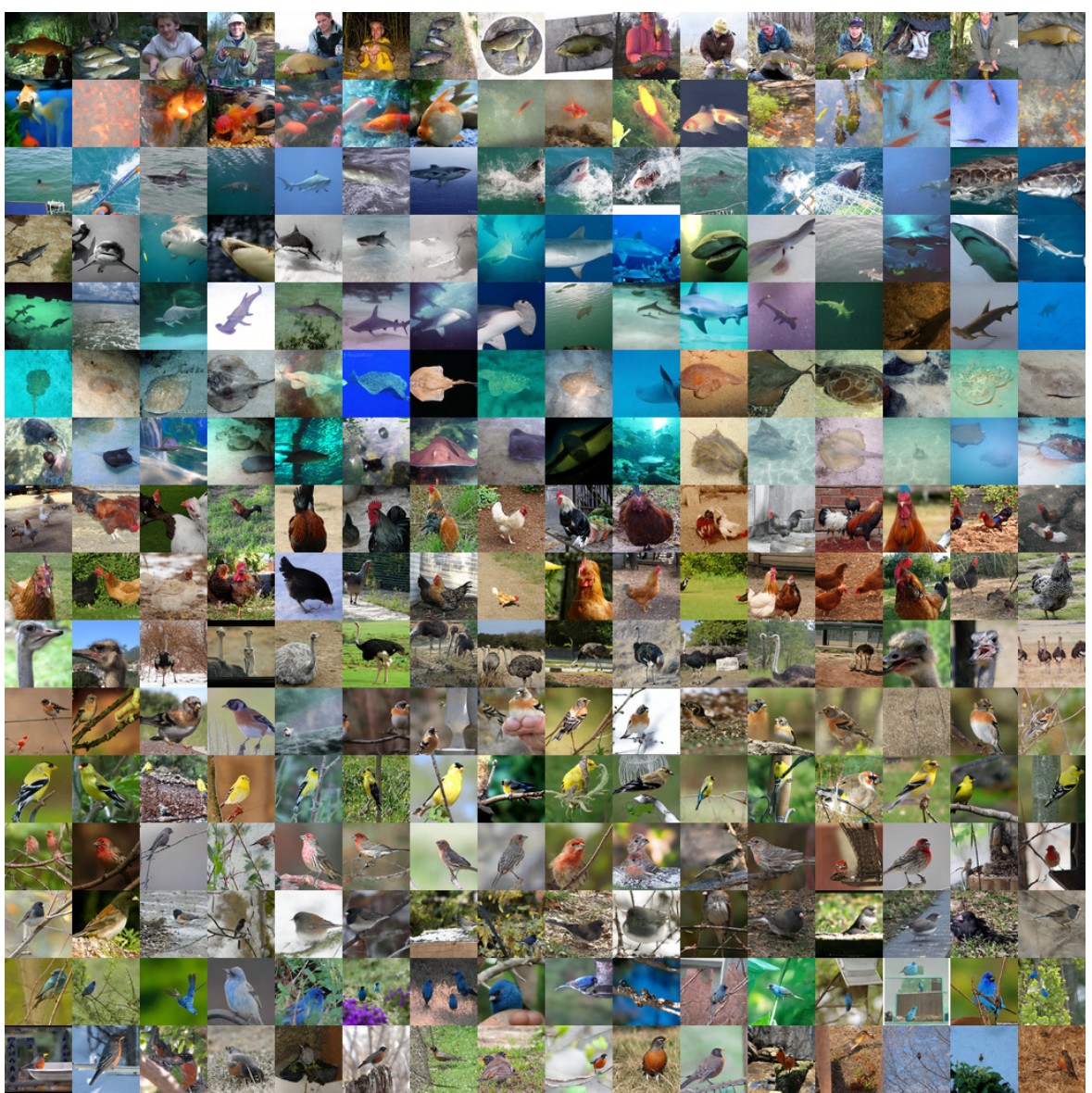

*Figure 5.* Conditionally generated $64 \times 64$ images on Imagenet using 2-step sampling.

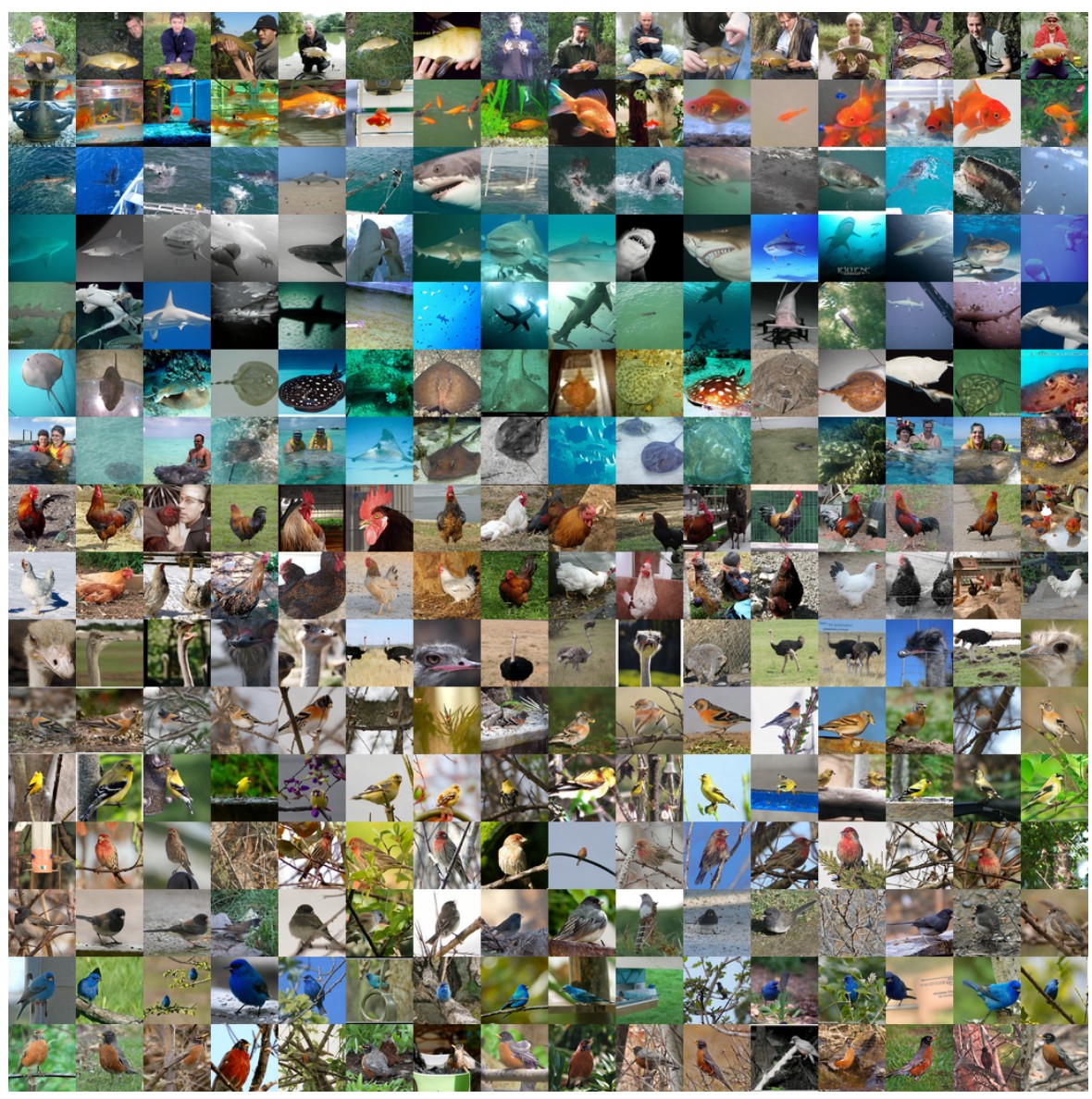

*Figure 6.* Conditionally generated $64 \times 64$ images on Imagenet using 4-step sampling.

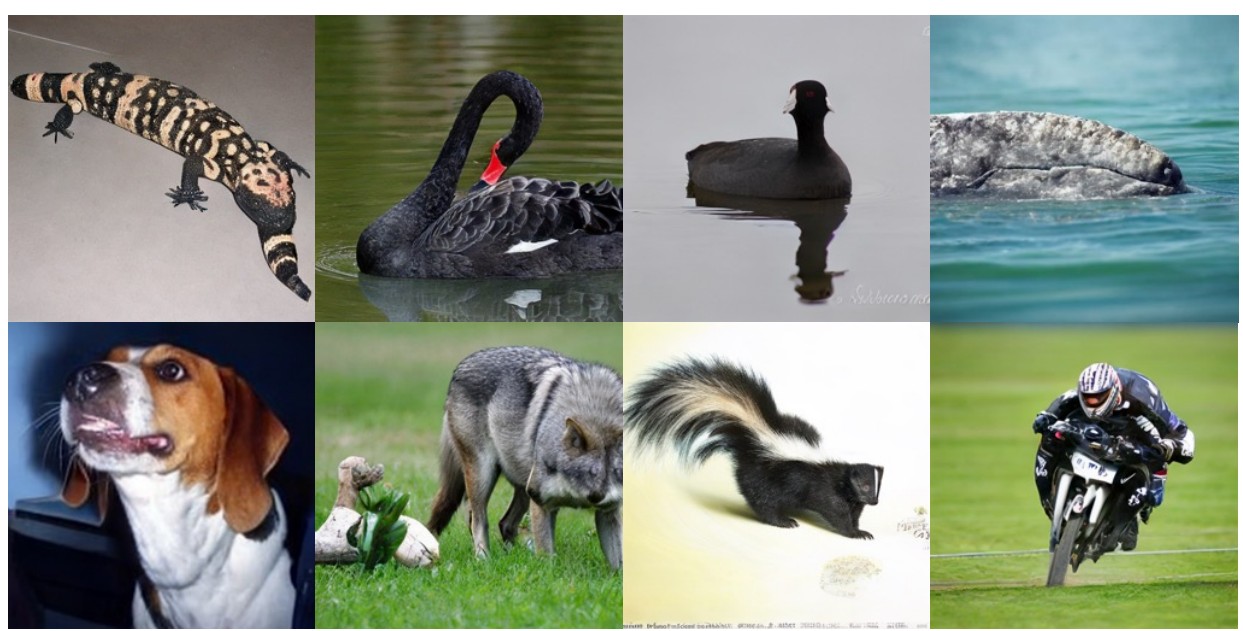

(a) 2 steps

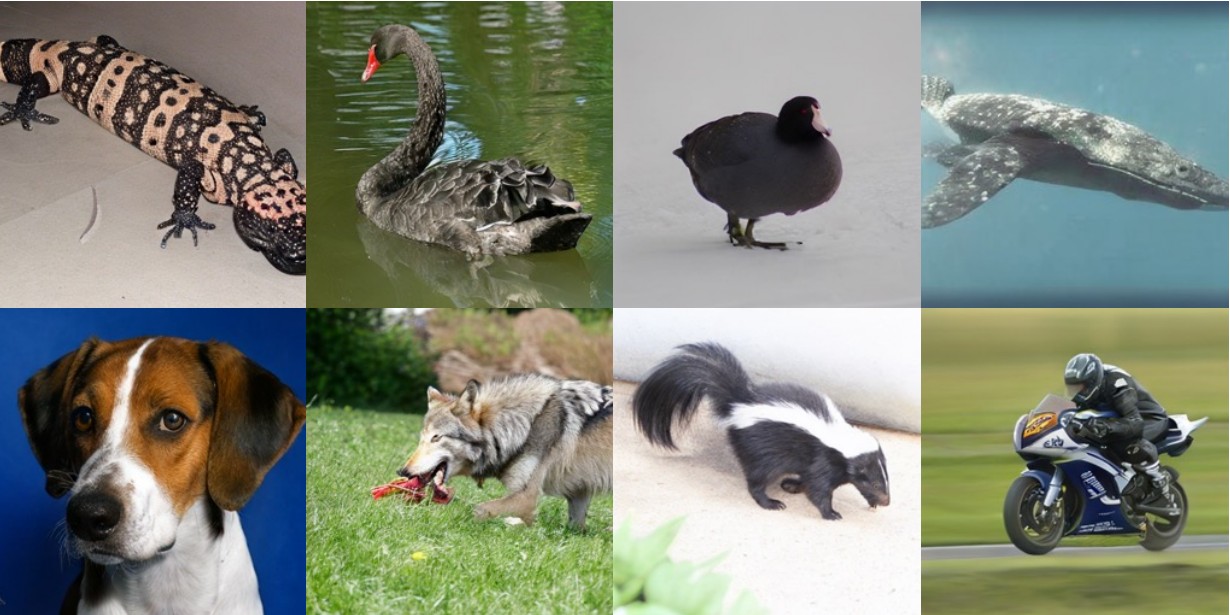

(b) 4 steps

*Figure 7.* Class-conditioned $512 \times 512$ images generated by CoSIM with different steps on Imagenet using M model, starting from identical noise.

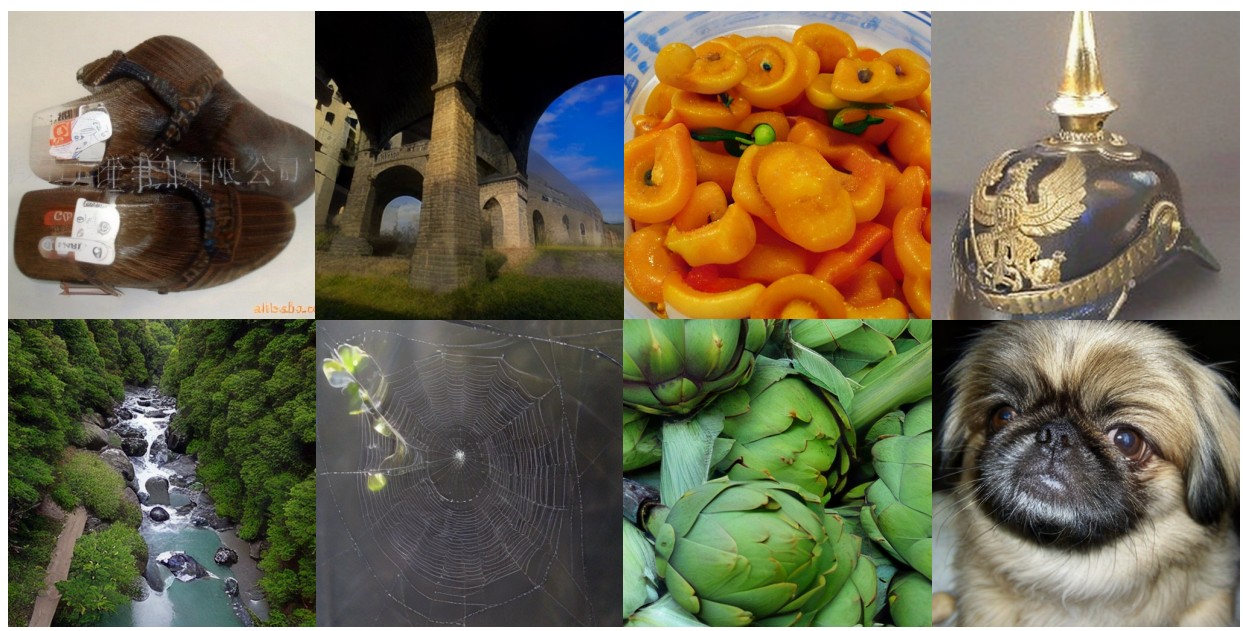

(a) 2 steps

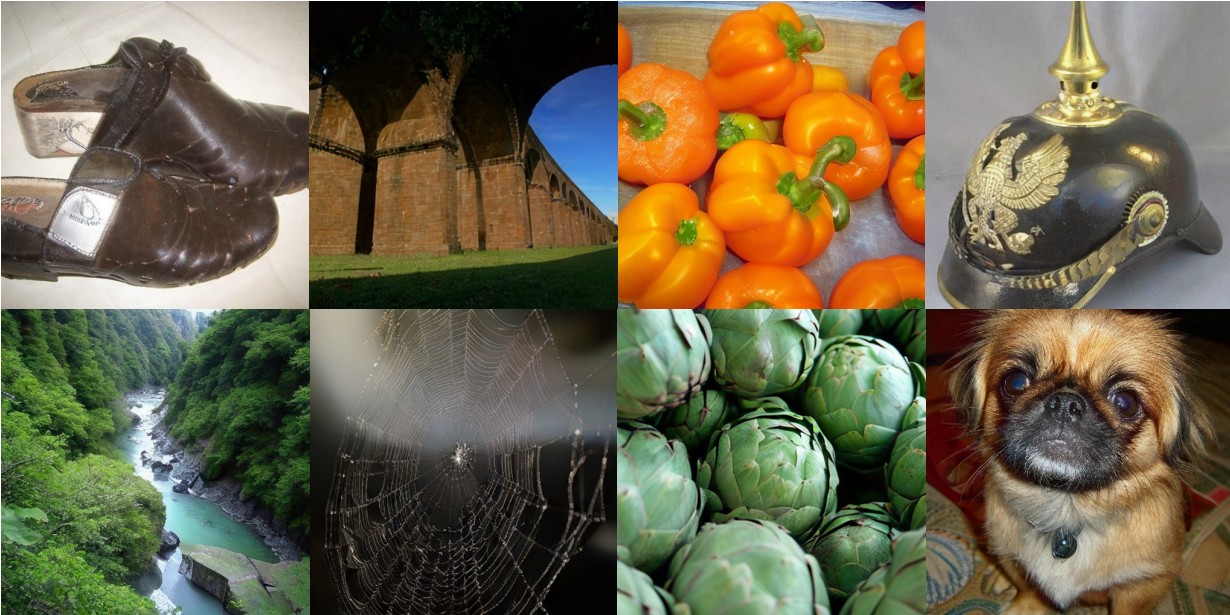

(b) 4 steps

*Figure 8.* Class-conditioned $512 \times 512$ images generated by CoSIM with different steps on Imagenet using L model, starting from identical noise.

