# OpenReview forum: "Continuous Semi-Implicit Models"
_ICML.cc/2025/Conference — ICML 2025 poster_

### Official Review · Reviewer_LH32 · 2025-03-10

**Overall Recommendation:** 1

**Summary:**

The paper studies semi-implicit models, and proposes an extension from hierarchical semi-implicit models (hSIM) to continuous-time SIMs (cSIM). The continuous-time SIM has the advantage of being simulation-free, as opposite to hierarchical SIM; and enables multi-step sampling at the same time. This is achieved thanks to a formulation inspired from consistency models. Based on this framework, the authors propose several adaptations to the hSIM framework, e.g. a shifted regularization term. The authors analyse some theoretical properties of their method, particularly the benefits of multi-step sampling. Finally, they demonstrate high performance on Cifar10, ImageNet 64x64, and ImageNet 512x512.

## update

I decide to not change my rating for the unclear link between the proposed method and consistency models:

The authors claim in the rebuttal that "our method surpasses ECT by a large margin on both FID and FD-DINOv2, details shown in Table 1 and Table 2 in paper.". However, this is not true when looking at fixed NFE. It surpasses on FD-DINOv2, but not in FID.
The claim that the proposed method allows for a faster training than consistency models would require more empirical evidence, by comparing both methods in a similar setting, with same number of iterations/compute budget. I am unsure that we can draw such conclusions.

**Claims And Evidence:**

The claims made in the paper are supported by empirical and theoretical evidence.
Theorems prove 1) the soundness of the shifted regularization term and of the training objective 2) the benefits of multi-step sampling. Empirical evidence show that the proposed method reaches SOTA-level performance on standard image benchmarks.

**Essential References Not Discussed:**

Not in my knowledge.

**Experimental Designs Or Analyses:**

I checked the experimental design and did detect any issue. The authors use standard benchmarks in the generative modelling literature (FID/FD on CIFAR-10 and ImageNet-64). An area of improvement would be to use more fine-grained evaluation metric, for example metrics that detect memorization and over-fitting in generative models (Stein et al. “Exposing flaws of generative model evaluation metrics and their unfair treatment of diffusion models”,  NeurIPS 2023).

**Methods And Evaluation Criteria:**

Method: The method is sound. The core idea is to use a consistency formulation in the setting of semi-implicit models. This resonates with ideas validated in recent works.

Evaluation criteria: The method is tested on standard image datasets (CIFAR-10, ImageNet) and with standard metrics (FID, FD-DINOv2).

**Other Comments Or Suggestions:**

No.

**Other Strengths And Weaknesses:**

Weakness 1: As already explained in Relation to Broader Literature, the extension proposed in this framework relies on the use of a "consistency" formulation, which is largely adopted in the generative modelling community. This limits the technical novelty of the paper.

Weakness 2:  To me, this paper asks the question: what is the best way to train a consistency model? Indeed, since your sampling is basically the same as the one of a standard consistency model, the main difference is in the training procedure. Do you have an opinion on why it would be better to rely on a Fisher divergence minimization rather than on a consistency loss? This is not clear since coSIM is not superior to ECT. There is no theoretical evidence, nor empirical evidence, that using a your training procedure leads to a better model.

Overall, the contribution is not clear to me. The proposed method seems to improve semi-implicit models with the consistent formulation. But are semi-implicit models needed for high-quality few-step generation? Why would this framework be more interesting than standard consistency models? This discussion is currently lacking in the paper.

**Questions For Authors:**

No.

**Relation To Broader Scientific Literature:**

This paper is studying the setting of semi-implicit models. It consists in applying an idea from consistency models (Song et al., 2023) to this setting. This allows to go from hierarchical semi-implicit models to continuous-time and simulation-free semi-implicit models. However, continuous-time SIM are very close to consistency models. Most notably, the sampling algorithm is the same. However, the training differs, since continuous-SIM are based on a Fisher divergence minimization scheme.

**Theoretical Claims:**

I did not verify in detail the theoretical proofs.

---

> ### Author Rebuttal · Authors · 2025-04-01
>
> Thank you for your valuable feedback! Below are our responses to your concerns.
>
> `Q1`: I checked the experimental design ... Stein et al. “Exposing flaws of generative model evaluation metrics and their unfair treatment of diffusion models”, NeurIPS 2023.
>
> `A1`: Exactly! The paper “Exposing flaws of generative model evaluation metrics and their unfair treatment of diffusion models” from NeurIPS 2023 by Stein et al, directly proposed the metric FD-DINOv2. They pointed out that FID as a metric is unfairly favoring GAN-style models, and has limited numeric accuracy to handle modern advanced diffusion models, as they usually reach a very small FID value. This inspires us to use FD-DINOv2 in our experiments.
>
> `Q2`: As already explained in Relation to Broader Literature ... limits the technical novelty of the paper.
>
> `A2`: Here we first clarify the novelty of our method.  CoSIM is derived from the framework of continuous semi-implicit variational inference. Methods such as HSIVI [1] and SiD [2] are built upon this semi-implicit training objective to accelerate diffusion models. Theoretically, We introduce a regularization framework of the SiD loss to stabilize the training and address the pathological behavior under large alpha values as demonstrated through theorem 3.1 and the ablation study on the regularization term coefficient $\mathrm{coef} = (1+\lambda) \alpha$.
>
> Table1: Class-conditional ImageNet (512x512) for CoSIM-L given **$\alpha=2.0$**.
> |$\mathrm{coef}$\Iteration | 204k| 1024k | 2048k | 4096k|
> |----|----|----|----| ----|
> |$\mathrm{coef}=0.5$ with NFE=$2$| 374.8 | 3960.0 | 4129.3 | 4039.9 |
> |$\mathrm{coef}=2.5$ with NFE=$2$| 619.0 | 114.1  | 78.7   | 64.9   |
>
> Furthermore, by using a specific parametrization, we derive a novel training method for consistency models at a distributional level.
> Unlike traditional consistency model fixed on the reversed ODE path,
> CoSIM learns the consistency map directly, without being constrained by intermediate process.
> Through our experiments, we show this achieves superior performance compared to the current state-of-the-art consistency models on ImageNet $512 \times 512$, using **much less training budget** (ours 20M compared to sCD [3] 819.2M).
> Table3: Sample quality measured by FID on class-conditional ImageNet (512x512).
> |Model|#params|\NFE|FID|
> |-|-|-|-|
> |sCD-S| 280M | 2 |2.50|
> |sCD-M| 498M | 2 |2.26|
> |sCD-L| 778M | 2 |2.04|
> |CoSIM-S| 280M| 2 | 2.66|
> |CoSIM-S| 280M| 4 | 2.56|
> |CoSIM-M| 498M|2| **1.95**|
> |CoSIM-M| 498M|4|1.93|
> |CoSIM-L| 778M|2|1.84|
> |CoSIM-L| 778M|4| **1.83**|
>
> [1] Zhou, et. al., "Score identity distillation: Exponentially fast distillation of pretrained diffusion models for one-step generation", ICML 2024.
>
> [2] Yu, et. al., Hierarchical Semi-Implicit Variational Inference with Application to Diffusion Model Acceleration, NeurIPS 2023
>
> [3] Lu, et al., "Simplifying, stabilizing and scaling continuous-time consistency models.", OpenAI
>
> `Q3`: Do you have an opinion ... Fisher divergence ... a better model.
>
> `A3`: As a novel training scheme derived from continuous semi-implicit variational inference, CoSIM differs fundamentally from traditional consistency models in the following aspects, offering distinct training advantages:
>
> - Fisher divergence does not require strictly following the reverse ODE trajectory but only learns a consistency mapping on a distributional level. This relaxes the choice on distillation path and enables various sampling strategies. It also reduces the training budget as the mapping can be more flexible. Conversely, ECT strictly follows the ODE trajectory, and its training budget is much higher than ours on comparable performance level.
> - Experimentally, our method surpasses ECT by a large margin on both FID and FD-DINOv2, details shown in Table 1 and Table 2 in paper.
> - Our approach primarily focuses on few-step sampling, which is also different from SiD, diff-instruct. We show that our approach shows significantly better performance using much less total training budget in our experiments, and scales to larger models better.
>
> `Q4`: Are semi-implicit models ... lacking in the paper.
>
> `A4`: To clarify, our contributions over semi-implicit models are:
> - We extend the framework of hierarchical semi-implicit variational inference models to the continuous settings.
> - We introduce a regularization term and give theoretical analysis on why this term could further stabilize the training. This addresses the pathological behavior of SiD loss when $\alpha$ is large.
> - Our formulation, under a specific parameterization, reforms to a consistency training on a distributional level. This enables a more flexible mapping rather than traditional consistency training on a fixed reverse ODE trajectory. This reduces training difficulty and reduces training budget, already validated by our experiments.
> - Please refer to A2 to see that our method can surpass the performance of latest consistency models with much lower training budget.

---

### Official Review · Reviewer_hsDu · 2025-03-12

**Overall Recommendation:** 3

**Summary:**

CoSIM is a continuous extension of hierarchical semi-implicit models, designed to enhance expressiveness and accelerate diffusion models with pretrained score networks. Unlike traditional hierarchical semi-implicit models, which suffer from slow convergence due to sequential training, CoSIM introduces a continuous transition kernel, enabling efficient, simulation-free training.

**Claims And Evidence:**

The authors claim that HSIVI suffers from slow convergence, while CoSIM's continuous framework improves training efficiency. It is recommended to provide empirical evidence supporting this claim, such as FID/FD vs. training iterations or runtime comparisons.

**Essential References Not Discussed:**

The authors discussed essential references.

**Experimental Designs Or Analyses:**

The overall validity of the experimental design appears valid and reasonable.

**Methods And Evaluation Criteria:**

The paper employs standard evaluation metrics, such as FID and FD-DINOv2, commonly used in image generation.

**Other Comments Or Suggestions:**

- Some notations are not defined or explained well:
    1. What is $q_t(x_t)$ in Lines 154–155, in the right-hand column?

    2. What is $L_{SIVI-f}$? Is it a symmetric divergence: it seems like  $L_{SIVI-f}(p_\text{parametrized}||p_\text{target})$ and $L_{\text{SIVI}-f}(p_\text{parametrized}||p_\text{target})$ are not consistent (Eq. (4), Eq. (9), and  [1])?

    3. What is $p_s(x_s)$ in Eq. (9)?



[1] Yu, et. al., Hierarchical Semi-Implicit Variational Inference with Application to Diffusion Model Acceleration


------

**After rebuttal:**
Although the extension to continuous time is relatively straightforward, I appreciate that the authors have included supplementary experiments to address my concerns regarding the empirical aspects. Thus, I raise my rating.

**Other Strengths And Weaknesses:**

- Is the training stable given that multiple networks are trained simultaneously?

- What is the numerical performance with a one-step generation?

- How does the parameter $\lambda$ affect performance? Is there an ablation study?

- How does CoSIM integrate with flexible (un-)conditional generation techniques, such as classifier-free guidance?

- How sensitive is the numerical performance to the choice of time distribution $\pi(\gamma | r_s)$? The same question applies to the weighting functions $w_i(s)$.

**Questions For Authors:**

Please see the comments above.

**Relation To Broader Scientific Literature:**

Extending the discrete-time framework (in HSIVI) to a continuous-time framework is natural, especially within the diffusion model community. However, the significant performance gain over HSIVI-SM suggests that this extension is both meaningful and interesting.

**Theoretical Claims:**

The theorems and proofs appear sound.

---

> ### Author Rebuttal · Authors · 2025-04-01
>
> Thank you for your valuable feedback! We will address your concerns below.
>
> `Q1`: It is recommended to provide empirical evidence supporting this claim, such as FID/FD vs. training iterations or runtime comparisons.
>
> `A1`: We give the details of our training iterations in Table 4 in Appendix C. For comparison with HSIVI, we give the following two tables.
>
> Table1: Training Efficiency and Performance on Unconditioanl CIFAR10
> |Method | \# of total training images| FID |
> |-|-|-|
> |HSIVI| 27 M | 4.17 (NFE=15)|
> |CoSIM| 20M | 2.55 (NFE=4)  |
> |CoSIM| 200M | 1.97 (NFE=4) |
>
> Table2: Comparison of Training Efficiency and Performance on ImageNet $64\times 64$
> |Method | \# of total training images| FID |
> |-|-|-|
> |HSIVI| 27 M | 15.49 (NFE=15)|
> |CoSIM| 20M  | 2.51 (NFE=4)|
> |CoSIM| 200M | 1.46 (NFE=4)|
>
> `Q2`: Is the training stable given that multiple networks are trained simultaneously? How does the parameter lambda affect performance? Is there an ablation study?
>
> `A2`: Yes, all networks are trained simultaneously and stably, which has also been observed in previous works such as SiD[1] and Diff-Instruct[2]. We provide an ablation on $\lambda$ in the form of coefficient $\mathrm{coef} := \alpha(1+\lambda)$  below.
>
> Table 3: Sample quality measured by FD-dinov2 on class-conditional ImageNet (512x512) for CoSIM-L given $\alpha=1.2$  and NFE=$2$.
> |$\mathrm{coef}$ \\# of total training images | 204k| 1024k | 2048k | 4096k|
> |---|----|----|----| ----|
> |$\mathrm{coef}=0.5$ | 309.7 | 101.3 | 69.6 | 61.5 |
> |$\mathrm{coef}=1.0$ | 392.8 | 92.8 | 61.7 | 50.5 |
> |$\mathrm{coef}=1.5$ | 421.9 | 95.2 | 58.6 | **49.2** |
>
> The results show that introducing regularization (i.e., increasing $\mathrm{coef}$) within a suitable range facilitates the learning of $f_\psi$, which in turn improves the training of $q_\phi$. Furthermore, since the reduction on FD-DINOv2 continues as training progresses, CoSIM demonstrates stable training across a relatively broader range of regularization coefficients, such as 1.0 and 1.5.
>
> [1] Zhou, et. al., "Score identity distillation: Exponentially fast distillation of pretrained diffusion models for one-step generation", ICML 2024
>
> [2] Luo, et. al., "Diff-instruct: A universal approach for transferring knowledge from pre-trained diffusion models", NeurIPS 2023
>
> `Q3`: The numerical performance with a one-step generation.
>
> `A3`:  CoSIM is inherantly a multi-step model, trained to model the continuous transition kernel rather than a one-step deterministic generation model like GANs. One significant advantage of multi-step generation is the ability to inject more randomness in each step, enhancing sampling diversity.
>
> For ablation, we also evaluated one-step generation quality of CoSIM on ImageNet $512 \times 512$ for M model.
>
> Table4: Sample quality on conditional ImageNet 512x512.
> |Model | \# params|NFE|FD-dinov2| FID |
> |-|-|-|-|-|
> |EDM2-M-dino| 498M |63 |58.44| - |
> |Moment Matching|400M |1| - |$3.3^*$ |
> |CoSIM-M| 498M| 1 | 74.23| 3.52|
>
> Note that FID result of Moment Matching[1] is evaluated at a lower resolution 128x128. Consider the difficulty of generating in higher resolution, our method is comparable with Moment Matching[1] on one-step generation.
>
> [1]Salimans, et. al., "Multistep distillation of diffusion models via moment matching." NeurIPS 2024.
>
> `Q4`: How does CoSIM integrate ... such as classifier-free guidance?
>
> `A4`: If an unconditional model $G_\phi(x_t; t,\emptyset)$ is used to modify the generation, the parameterization can be considered to integrate it into CoSIM's training.
> $$
> G_\phi^{(w)}(x_t; t,c) = wG_\phi(x_t; t,c) + (1-w)G_\phi(x_t; t,\emptyset).
> $$
>
> We also noted that SiD-LSG [1] proposed a novel guidance method by using CFG in training and inferencing with different weights $w$. This method also adapts to our formulation and provides inspiration for future work.
>
> [1] Zhou, et. al., "Long and short guidance in score identity distillation for one-step text-to-image generation."
>
> `Q5`: The numerical performance to the choice of time distribution? ... the weighting functions.
>
> `A5`: We give another ablation study on the hyperparameter $R$ in the time distribution $\pi(\gamma|r_s)$.
>
> Table5: Ablation on $R$ for CoSIM-M on class-conditional ImageNet ($512 \times 512$) with NFE=4 by FD-DINOv2
> |$R$  \\# of total training images | 2M | 4M| 38M |
> |-|-|-|-|
> |$R=4$| 220.2 | 73.6 | 57.8 |
> |$R=6$| 110.4 | 55.6 | 54.7|
> |$R=8$| 95.6 | 54.7 | 55.8|
>
> Different $R$ gives slightly different convergence rate during early stages of training.
> But after 38M training images, this difference is minimized.
>
> For the weighting functions, we adopt the same weighting function as in EDM[1], this setting has been widely adopted in many works such as SiD and irrelevant to our algorithm design. Hence we keep it fixed.
>
> [1] Karras, et. al., "Elucidating the Design Space of Diffusion-Based Generative Models", NeurIPS 2022
>
> `Q6`: Typos
>
> `A6`: Thank you for catching these typos! We will correct them in our revision.

---

### Official Review · Reviewer_yUwN · 2025-03-25

**Overall Recommendation:** 3

**Summary:**

The paper proposes Continuous Semi-Implicit Models (CoSIM), extending hierarchical semi-implicit models into a continuous-time framework for diffusion model acceleration.
CoSIM introduces a continuous transition kernel that allows the simulation-free training.
It uses semi-implicit variational inference (SIVI) as training criteria, providing a distributional-level consistency model without relying on moment-level or sample-level reverse processes.
The method yields comparable or superior results on FID and improves existing approaches on the FD-DINOv2 metric on CIFAR-10 and ImageNet datasets.
The authors further provide theoretical insights into their approach via error analysis.

**Claims And Evidence:**

The paper makes the following claims:

- Faster Convergence: The authors show that CoSIM reduced the number of iterations required for distillation.

- Improved Generation Quality: CoSIM achieves comparable or better generation quality. Supported by Comparable FIDs and better FD-DINOv2.

- Continuous-time Framework: The authors extend the previous SIVI to a continuous framework.

**Essential References Not Discussed:**

NA

**Ethical Review Flag:**

Flag this paper for an ethics review.

**Experimental Designs Or Analyses:**

- Standard benchmarks on CIFAR-10 and ImageNet datasets for unconditional and conditional generation tasks, respectively.
- Comparison against strong, established baselines such as SiD, iCT, ECT, CTM, and Moment Matching methods.
- Reporting quantitative (FID, FD-DINOv2) and qualitative evaluations (sample images).

Experiments look valid and sound. This paper also outlines experiment setups and hyperparameters.

**Methods And Evaluation Criteria:**

- This paper evaluates CoSIM using widely accepted metrics (FID, FD-DINOv2) and datasets (CIFAR-10, ImageNet) for generative modeling.
- The FD-DINOv2 metric specifically helps evaluate perceptual alignment with human vision, a useful complement to standard FID metrics.

The chosen evaluation makes sense given the objective of achieving accelerated image generation.

**Other Comments Or Suggestions:**

I personally like this paper. It extends HSIVI to continuous-time framework and provides a solid theorectical ground for prior works like SiD and validates the regularization techniques.

However, I find the paper's emphasis on its connection to consistency models overstated. The current approach seems more naturally framed as an extension of HSIVI to continuous-time settings or as a multi-step generalization of SiD. The relation to consistency models feels unnecessary and shouldn't be demonstrated as the paper's contributions.

**Other Strengths And Weaknesses:**

NA

**Questions For Authors:**

- Regularization Strength:  How sensitive is CoSIM to the choice of the regularization hyperparameter?

- Computational Overhead: Could you clarify or quantify the computational overhead introduced compared to simpler baseline distillation methods, such as SiD and Diff Instruct?

**Relation To Broader Scientific Literature:**

- Builds upon recent advances in diffusion model acceleration and semi-implicit variational inference (SIVI).
- Identifies the advantages of continuous over discrete-time frameworks in hierarchical generative modeling.
- Compares itself to recent notable works (e.g., SiD, HSIVI), effectively highlighting contributions and improvements.

**Theoretical Claims:**

The paper provides several theoretical claims:

- Derivation of the continuous transition kernel.
- Proofs of unbiasedness and error bounds for two-stage optimization (Theorem 3.1).
- Error analysis (Propositions 3.3 and 3.4), providing theoretical guarantee about the quality of multi-step sampling.

---

> ### Author Rebuttal · Authors · 2025-04-01
>
> Thank you for providing valuable feedback! Here are our responses.
>
> `Q1`: I personally like this paper. It extends HSIVI ... paper's contributions.
>
> `A1`: Thank you for your interest in our work! We acknowledge that CoSIM can be regarded as a continuous extension of HSIVI, but our contributions go beyond that and is different from traditional consistency models in the following aspects:
> - Unlike traditional ODE trajectory distillation applied by normal consistency distillation methods, we train the consistency model at a distributional level using Fisher divergence. This approach eliminates the need to constrain the generator $G_\phi$ to sample along a reversed ODE path predefined by the score function, allowing the model to learn a more flexible consistency mapping on a distributional level.
> - Experimentally, we find that this distinction enables CoSIM to train much faster outperform the state-of-the-art consistency model sCD [1], using only 20M total training images (compared to the 819.2M by sCD). We provide extended results of CoSIM compared with sCD on FID, demonstrating its superior performance across various model sizes.
>
> Table1: Sample quality measured by FID on class-conditional ImageNet (512x512).
> |Model|#params|\NFE|FID|
> |----|----|----|----|
> |EDM2-S-FID| 280M| 63 | 2.56|
> |EDM2-M-FID| 498M| 63 | 2.25|
> |EDM2-L-FID| 778M| 63 | 1.96|
> |sCD-S| 280M | 2 |2.50|
> |sCD-M| 498M | 2 |2.26|
> |sCD-L| 778M | 2 |2.04|
> |CoSIM-S| 280M| 2 | 2.66|
> |CoSIM-S| 280M| 4 | 2.56|
> |CoSIM-M| 498M| 2 | **1.95**|
> |CoSIM-M| 498M| 4 | 1.93|
> |CoSIM-L| 778M| 2 | 1.84|
> |CoSIM-L| 778M| 4 | **1.83**|
>
> `Q2`: How sensitive is CoSIM to the choice of the regularization hyperparameter?
>
> `A2`: We conducted an ablation study on $\lambda$ with the coefficient $\mathrm{coef} := \alpha(1+\lambda)$ in (15) for conditional ImageNet $512 \times 512$ generation of CoSIM using the $L$ model size, with fixed $\alpha=1.2$.
> The results demonstrate that introducing regularization (i.e., increasing $\mathrm{coef}$) significantly enhances the learning process of $f_\psi$ in a wide range.
>
> Table2: Sample quality measured by FD-dinov2 on class-conditional ImageNet (512x512) for CoSIM-L given $\alpha=1.2$.
> |$\mathrm{coef}$\ \# of total training images  | 204k| 1024k | 2048k | 4096k|
> |-|-|-|-|-|
> |$\mathrm{coef}=0.5$ with NFE=$2$| 309.7 | 101.3 | 69.6 | 61.5 |
> |$\mathrm{coef}=1.0$ with NFE=$2$| 392.8 | 92.8 | 61.7 | 50.5 |
> |$\mathrm{coef}=1.5$ with NFE=$2$| 421.9 | 95.2 | 58.6 | 49.2 |
>
> `Q3`: Computational Overhead: Could you clarify or quantify the computational overhead ... SiD and Diff Instruct?
>
> `A3`: There are three networks in the training process, generator $G_\phi$, teacher $S_{\theta^*}$ and auxiliary function $f_{\psi}$. During training, teacher $S_{\theta^*}$ is frozen while generator $G_\phi$ and auxiliary function $f_{\psi}$ are simultaneously evolving.  For SiD and Diff Instruct, all three networks are kept the same as the baseline methods and initialized from the same checkpoint. For our method, the auxiliary function $f_{\psi}$ is modified to incorporate the extra time parameter $s$ introduced in Section 3. In practice, we duplicate the time embedding layer, which only adds a very small number of extra parameters.
>
> Specifically, on CIFAR, the total number of trainable parameters is 111,465,478 for SiD and Diff-Instruct. With our modification, it increases to 116,128,006, adding only 4% extra parameters.
> On ImageNet $64\times64$, the total number of trainable parameters is 59,1798,534 for SiD and Diff-Instruct. With our modification, it increases to 619,999,878, also adding only 4% extra parameters.
>
> Additionally we time the computational overhead for SiD, Diff-Instruct and our method on unconditional CIFAR generation. We conduct all experiments on a single L40S GPU with all fixed hyperparameters such as batch size, learning rate, optimizer etc..
>
> Table 3: training time for 10496 images
> | Time \ Model|   SiD   | Diff-Instruct |  CoSIM  |
> |------|:-------:|:-------------:|:-------:|
> | Time | 2min28s |    2min42s    | 3min08s |
>
> Compared to SiD and Diff-Instruct, our method requires 27% and 18% more training time respectively. Apart from the extra trainable parameters explained above, the regularization we introduced in Theorem 3.1 also contributes to the extra time.
>
> We argue such overhead is worthy, as the overall training objective stabilizes and the total training budget shrinks. On conditional imagenet $64\times64$ generation, our method only requires 200M training images to reach FID 1.46, surpassing SiD which needs 1000M training images to reach FID 1.52. On the same task, Diff-Instruct reaches FID 5.57 with 4.8M training images. Since we aim for a much lower FID, we did not frequently test our model in the early stages of training. The closest one we have is that our model reaches FID 4.15 with 5.12M training images.

---

> > ### Comment · Reviewer_yUwN · 2025-04-06
> >
> > I appreciate the authors' response, including the additional experimental results, clarification on the method, and the ablation on the regularization strength.
> >
> > That said, I must **reiterate my concern** regarding the connection to consistency models. In my view, this connection is both unnecessary and overstated. CoSIM is much more naturally situated within the lineage of score distillation methods starting from SDS, such as SiD, VSD, and Diff-Instruct. Like these methods, CoSIM relies on a teacher model and two evolving networks to perform distribution-level matching.
> >
> > What CoSIM brings to this family is valuable: it extends HSIVI to modern generative models with strong performance, improves the stability of these approaches, and introduces a meaningful regularization mechanism.  However, I see 0% alignment with consistency models. Framing CoSIM as a consistency model not only adds confusion but also detracts from the strength of the actual contribution.
> >
> > While I **initially leaned positively** toward this work, the authors' insistence on the framing has led me to **shift to a more neutral stance**. My recommendation for acceptance now depends on a substantial revision of the paper that repositions CoSIM clearly within the context of score distillation and SIVI, and removes the overclaims about consistency models. Without such a revision, I would not support acceptance.
> >
> > To be clear, my concern is **not with the method itself** or the theoretical foundation, but with **the framing and claims** made in the paper. I am very confident in this assessment.

---

> > > ### Author Response · Authors · 2025-04-06
> > >
> > > Thank you for taking the time to carefully review our work and for pointing this out.
> > > We acknowledge that we have not clearly distinguished CoSIM from consistency models.
> > > CoSIM utilizes a parameterization approach that shares some similarities with consistency distillation, leveraging the consistency map $G_\phi(x_t, t)$.
> > > However, its methodology is **fundamentally rooted in the framework of score distillation**, which is notably distinct from the ODE trajectory distillation used in consistency models.
> > >
> > > To address this, we will focus in Sections 3.1 and 3.2 on presenting our continuous HSIVI framework, situating it within the context of score distillation methods such as SiD.
> > > **In Section 3.3, we will revise the title** "A Path Towards Consistency" to "Parameterization of CoSIM."
> > >
> > > Additionally, we will **update the abstract and conclusion** by replacing the phrase "provides a novel method of consistency model training on a distributional level" with "we provide a novel method for multistep distillation of generative models training on a distributional level."
> > > We will also **revise the introduction** by replacing the phrase "The design of the continuous transition kernel aligns CoSIM closely with consistency models" with "The design of the continuous transition kernel shares some similarities with consistency distillation (CD)."
> > >
> > > Thanks again for identifying this ambiguous claim. We will make these modifications in our revision.

---

### Official Review · Reviewer_vhoo · 2025-03-25

**Overall Recommendation:** 3

**Summary:**

This paper proposes CoSIM a continuous hierarchical variational inference framework for semi-implicit distribution aiming to accelerate the sampling speed of diffusion models. The main contribution of this paper includes:

 * The authors propose a score-based diffusion distillation approach demonstrating superior performance on DINOv2 metrics comparing to data-free diffusion distillation algorithms with 4 sampling steps across multiple datasets (CIFAR-10, ImageNet 64×64, and ImageNet 512×512).
 * The authors theoretically connects the Hierarchical Semi-Implicit Variational Inference (HSIVI) frame work with the Score-identity Distillation (SiD) and propose a regularization technique to address the bias of the fake score network in the score matching loss.
 *  The authors demonstrate that the generator network of SiD is intrinsically a consistency function and prove that a few-step sampler could potentially be a sweet spot in the tradeoff between iterative refinement sampling and error accumulation in multi-step sampling

## update after rebuttal: See the comments below

**Claims And Evidence:**

This paper overall supports the claims with clear evidence. However, my major concerns lies in the following two aspects:

* The authors mentioned in line 199-202 that when $\nabla \log q_{\phi} (x_s;s,t)$ deviates significantly from the target score model $S_{\theta}(x_s;s)$, this initialization strategy becomes inefficient for the second-stage optimization in equation(10). It does not seem very clear to me what is the bias in the SiD loss suggest by the authors and how this is connected to the regularization, I would appreciate more elaboration from the authors on this, either theoretically or through ablation studies on $\lambda$.

* The authors mentioned in line 167-169 that the fused loss function exhibits pathological behavior when $\alpha>1$, I guess this is a typo? Otherwise it would be interesting to ask why the authors follows $\alpha = 1.2$ in the implementation of CoSIM. Given the combination of $\lambda$ and $\alpha$, it would also be interesting to see the ablation studies on $alpha$ if time permits.

**Essential References Not Discussed:**

Related works that are essential to understanding the key contributions of the paper are cited.

**Experimental Designs Or Analyses:**

I noticed that CoSIM 4-step differs by only 0.06 from SiD on ImageNet 64×64, yet shows a significant 20-point improvement in FID DINOv2. Interestingly, CoSIM 2-step differs by a larger margin of 1.83 from SiD on ImageNet 64×64, while maintaining the same 20-point improvement in FID DINOv2. I would appreciate intuition on this observation.

**Methods And Evaluation Criteria:**

The methods and evaluation make sense to me.

**Other Comments Or Suggestions:**

Please see the above sections.

**Other Strengths And Weaknesses:**

The paper provides solid theoretical derivation backed with extensive experiment. However, I think the Section 3.1 and 3.2 essentially re-derives the SiD loss for a few-step sampler, which is limited the novelty. Though the derivation through Legendre-Fenchel Dual provide insights on the loss function of SiD, the improvement comparing to the SiD bi-level algorithm optimization is limited. The derivation on the error bounds for multi-step sampling could provide insights on few-step diffusion acceleration. The contribution of this paper is solid yet the overall significance and novelty is moderate.

**Questions For Authors:**

Please see the Claims And Evidence section. It would be helpful if the authors could elaborate more on the $f_{\psi}$ shifting.

**Relation To Broader Scientific Literature:**

The key contributions of this paper can be related to accelerating the inference of diffusion models and enriched the tools in hierarchical variational inference and provide a novel perspective on Score-identity distillation for diffusion models.

**Theoretical Claims:**

The proof for Theorem B1 Proposition 3.2 and 3.4 make sense to me. My major concern lies in the statement for Proposition 3.4 and assumption 3.3, $\epsilon_f$ may be inflated here due to the regularization, so it would be good to show that how regularization benefits the training of CoSIM.

---

> ### Author Rebuttal · Authors · 2025-04-01
>
> Thank you for your thoughtful review and valuable feedback. We address your specific questions and comments below.
>
> `Q1`: It does not seem very clear to me what is the bias in the SiD loss ... on $\lambda$.
>
> `A1`: Thank you for your valuable suggestion! The significance of our regularization comes in three aspects:
> - Regularizing the second-stage optimization of (10) (as discussed in Theorem 3.1) helps guide the optimal $f\_{\tilde{\psi}^*(\phi)}$ towards the pretrained score function $S\_{\theta^*}(\cdot;s) \\approx p(\cdot;s)$. And the Nash equilibrium of $\\phi^*$ remains consistent as in Theorem 3.1.
> - Our formulation strictly contains SiD loss as a special case, which is equivalent to $\alpha = 1.2$ and $\alpha(1+\lambda) = 0.5$. This introduces a potential bias when $f_\psi$ is perfectly trained and $\alpha$ is too large.
> As shown in (38), this causes $q_\phi$ to optimize in a direction that deviates from the target distribution, corresponding to the failure case in Figure 7 of SiD.
> Our formulation allows for a larger $\lambda$, but still ensuring positive coefficient $(1+2\lambda)$ in (38), mitigating the potential bias in SiD loss.
> - The ablation study on $\mathrm{coef} := \alpha(1+\lambda)$ in (15) with $\alpha=1.2$ show that introducing regularization facilitates the learning of $f_\psi$, improving the training of $q_\phi$.
>
> Table1: Class-conditional ImageNet (512x512) for CoSIM-L given **$\alpha=1.2$**.
> |$\mathrm{coef}$ \\# of total training images  | 204k| 1024k | 2048k | 4096k|
> |-|-|-|-|-|
> |$\mathrm{coef}=0.5$ with NFE=$2$| 309.7 | 101.3 | 69.6 | 61.5 |
> |$\mathrm{coef}=1.5$ with NFE=$2$| 421.9 | 95.2 | 58.6 | 49.2 |
>
>
> `Q2`: The authors mentioned in line 167-169 that the fused loss function ... on $\alpha$ if time permits.
>
> `A2`: The pathological behavior when $\alpha > 1$ refers to the SiD loss which is equivalent to $\mathrm{coef} := \alpha(1+\lambda) = 0.5$. We propose using a regularization term by increasing $\lambda$ to mitigate this issue, as discussed in A1.
>
> The ablation study below with a larger $\alpha=2.0$ shows the effectiveness of our regularization and the pathological results of SiD loss is similar to the observations shown in Figure 7 in SiD's paper.
>
> Table2: Class-conditional ImageNet (512x512) for CoSIM-L given **$\alpha=2.0$**.
> |$\mathrm{coef}$\Iteration | 204k| 1024k | 2048k | 4096k|
> |-|-|-|-|-|
> |$\mathrm{coef}=0.5$ with NFE=$2$| 374.8 | 3960.0 | 4129.3 | 4039.9 |
> |$\mathrm{coef}=2.5$ with NFE=$2$| 619.0 | 114.1  | 78.7   | 64.9   |
>
> `Q3`: My major concern lies in the statement for Proposition 3.4 and assumption 3.3 ... regularization.
>
> `A3`: $\epsilon_f$ measures the gap between $f_\psi$ and its optimal $f_{\psi^*}$ in the second optimization problem described in (12). This optimization is a quadratic problem, and in the optimal case, $\epsilon_f$ approaches 0. Therefore, the introduction of regularization does not affect $\epsilon_f$'s convergence to 0. Moreover, when $f_\psi$ is initialized as $S_\theta$, regularization brings $f_\psi$ closer to the optimal $f_{\psi^*}$, as shown in (16).
>
> `Q4`: I noticed that CoSIM 4-step differs by only 0.06 ... observation.
>
> `A4`: FID and FD-DINOv2 are designed in different numerical scales. FD-DINOv2 was designed more suitably for diffusion model evaluation. Results below are some well-known diffusion models evaluated for conditional image synthesis on ImageNet $256\times 256$.
> | Metric \ Model|   ADM  |   LDM  |
> |-|-|-|
> | FID       |  4.59  |  3.60  |
> | FD-DINOv2 | 203.45 | 112.40 |
>
> Furthermore, FID was introduced in 2017 and does not have enough numerical range to differentiate strong modern diffusion models. FD-DINOv2 was introduced to provide a larger numerical gap to solve this issue.
>
> `Q5`: The contribution of this paper is solid yet the overall significance and novelty is moderate.
>
> `A5`: Thank you for acknowledging the novelty of theoretical derivation of the fused loss and insight of regularization.
> We believe that CoSIM contributions go beyond SiD in the following aspects:
>
> - We theoretically introduce a regularization framework of the SiD loss to stabilize the training and solve the pathological behavior under large alpha values.
> - We propose a continuous HSIVI training scheme. With a specific parametrization, we derive a novel training method for consistency models at a distributional level.
> - Experimentally, our method performs better than the state-of-the-art consistency model sCD [1], using much less training budget (ours 20M compared to sCD 819.2M).
>
> Table3: Sample quality measured by FID on class-conditional ImageNet (512x512).
> |Model|#params|NFE|FID|
> |-|-|-|-|
> |sCD-S| 280M | 2 |2.50|
> |sCD-M| 498M | 2 |2.26|
> |sCD-L| 778M | 2 |2.04|
> |CoSIM-S| 280M| 2 | 2.66|
> |CoSIM-S| 280M| 4 | 2.56|
> |CoSIM-M| 498M| 2 | **1.95**|
> |CoSIM-M| 498M| 4 | 1.93|
> |CoSIM-L| 778M| 2 | 1.84|
> |CoSIM-L| 778M| 4 | **1.83**|
>
> [1] Lu Cheng et al. "Simplifying, stabilizing and scaling continuous-time consistency models."

---

### Decision · Program_Chairs · 2025-05-01

**Decision:**

Accept (poster)

**Comment:**

The paper unifies Hierarchical Semi-Implicit Variational Inference (H-SIVI) and Score identity Distillation (SiD) to construct a scaled Fisher divergence. Minimizing this divergence is achieved through an alternating optimization scheme that switches between optimizing a generator (given the score network) and optimizing the fake score network (given the generator).  While the generator loss resembles that of SiD, it diverges from SiD, which targets one-step generation, by optimizing for multi-step generation. Moreover, an additional regularization term, controlled by a tunable parameter $\lambda$, is introduced into the loss of the fake score network.

The paper refers to this continuous-time optimization framework as Continuous Semi-Implicit Models (CoSIM). CoSIM enables the same model to be flexibly applied across different numbers of sampling steps, as opposed to training a separate model for a fixed number of inference steps. Empirical results demonstrate improved performance when increasing the number of steps from 2 to 4, with the 4-step generation achieving competitive FID scores and SoTA results in FD-DINO_v2. These findings are supported by both detailed theoretical analysis and convincing experimental results on CIFAR10, ImageNet 64x64, and ImageNet 512x512.

The AC considers the paper above the acceptance threshold given the amount of original theoretical and methodological contributions. However, the AC recommends that the authors revise the paper to better position the paper in the literature:

A primary concern raised by reviewers is the paper's positioning of CoSIM relative to consistency models, and the perception that it attempts to present CoSIM as a superior alternative. While the concern of one reviewer was addressed by the authors’ promise to revise the description of consistency models, another reviewer remained cautious regarding this comparison. The final scores remained at 3-3-3-1. The AC found the authors’ rebuttal, particularly the comparison between sCD and CoSIM, sufficient to alleviate these concerns and opted to follow the majority recommendation to accept the paper.

The AC agrees with the authors’ plan to revise the paper to better position CoSIM in relation to consistency models. In addition, based on their own reading of the submission, the AC suggests the following additional revisions:



1. Report CoSIM's one-step generation results. While CoSIM is designed for arbitrary-step inference and may underperform in the one-step setting compared to methods tailored for it, including these results would provide a more comprehensive understanding of CoSIM’s strengths and limitations.

2. Report CoSIM's generation results beyond 4 steps. This will help understand how much room is left for improvement when the number of inference steps is further increased.

3. Add FID comparisons reported in the rebuttal, and add comparisons with SiD on ImageNet 512×512. While the original SiD paper did not include results on ImageNet 512x512, the FID and FD-DINO_v2 results of SiD were later reported in: "Zhou, M., Zheng, H., Gu, Y., Wang, Z., & Huang, H. (2025). Adversarial Score Identity Distillation: Rapidly Surpassing the Teacher in One Step. ICLR 2025." Although per ICML reviewing guidelines, this paper was not considered in the decision-making process, its reported results offer useful reference points for evaluating CoSIM against previous baselines on ImageNet 512×512.